# Discovery of a dual Ras and ARF6 inhibitor from a GPCR endocytosis screen

Jenna Giubilaro [1,2], Doris A. Schuetz [3], Tomasz M. Stepniewski[4,5], Yoon Namkung [2,6], Etienne Khoury[6], Mónica Lara-Márquez[2,7], Shirley Campbell[8], Alexandre Beautrait[3,10], Sylvain Armando[6], Olivier Radresa[6], Jean Duchaine[3], Nathalie Lamarche-Vane [2,7], Audrey Claing[8], Jana Selent[4], Michel Bouvier [3,9], Anne Marinier [3] & Stéphane A. Laporte [1,2,6✉]

Internalization and intracellular trafficking of G protein-coupled receptors (GPCRs) play pivotal roles in cell responsiveness. Dysregulation in receptor trafficking can lead to aberrant signaling and cell behavior. Here, using an endosomal BRET-based assay in a high-throughput screen with the prototypical GPCR angiotensin II type 1 receptor (AT1R), we sought to identify receptor trafficking inhibitors from a library of ~115,000 small molecules. We identified a novel dual Ras and ARF6 inhibitor, which we named Rasarfin, that blocks agonist-mediated internalization of AT1R and other GPCRs. Rasarfin also potently inhibits agonist-induced ERK1/2 signaling by GPCRs, and MAPK and Akt signaling by EGFR, as well as prevents cancer cell proliferation. In silico modeling and in vitro studies reveal a unique binding modality of Rasarfin within the SOS-binding domain of Ras. Our findings unveil a class of dual small G protein inhibitors for receptor trafficking and signaling, useful for the inhibition of oncogenic cellular responses.

---

[1] Department of Pharmacology and Therapeutics, McGill University, Montréal, QC, Canada. [2] Research Institute of the McGill University Health Center (RI-MUHC), Montreal, QC, Canada. [3] Institute for Research in Immunology and Cancer (IRIC), Université de Montréal, Montréal, QC, Canada. [4] Research Programme on Biomedical Informatics (GRIB), Department of Experimental and Health Sciences of Pompeu, Fabra University (UPF)-Hospital del Mar Medical Research Institute (IMIM), Barcelona, Spain. [5] InterAx Biotech AG, Villigen, Switzerland. [6] Department of Medicine, Research Institute of the McGill University Health Center (RI-MUHC), McGill University, Montréal, QC, Canada. [7] Department of Anatomy and Cell Biology, McGill University, Montréal, QC, Canada. [8] Department of Pharmacology and Physiology, Université de Montréal, Montréal, QC, Canada. [9] Department of Biochemistry and Molecular Medicine, Université de Montréal, Montréal, QC, Canada. [10] Present address: Schrödinger, Inc., New York, NY, United States. ✉email: stephane.laporte@mcgill.ca

Cell surface internalization of hormone receptors, such as G protein-coupled receptors (GPCRs) and receptor tyrosine kinases (RTKs), as well as their trafficking into endosomal compartments are fundamental to maintain cell responsiveness and homeostasis through the spatial and temporal regulation of signals[1–3]. Dysregulation of receptor trafficking and signaling such as for the RTK epidermal growth factor receptor (EGFR) has been well documented in the development of cancer[3]. Although unregulated GPCR signaling has been known to drive oncogenic responses[4], recent evidence also suggests that receptor internalization and/or interactions with proteins involved in regulating their trafficking also contribute to the initiation and progression of cancer for some GPCRs (for review see refs. [1,5,6]). Moreover, the interplay between these processes has also been implicated in the progression of cancer[7,8].

GPCRs, such as the prototypical receptors angiotensin II type 1, bradykinin B2, β2-adrenergic receptors, and vasopressin V2 receptors (AT1R, B2R, β2AR and V2R, respectively), signal at the plasma membrane (PM) mainly via the activation of their cognate G proteins (e.g., $G_{\alpha q}$ for AT1R and B2R, and $G_{\alpha s}$ for β2AR and V2R), leading to the activation of many downstream kinases, including mitogen-activated protein kinases (MAPK) like ERK1/2[9–11]. G protein-dependent activation of ERK1/2 by GPCRs also involves many pathways and effectors, and in many cases the activation of multiple G protein subtypes by the same receptor. For instance, $G_{\alpha s}$-coupled GPCRs activate MAPK through cAMP-dependent protein kinase A (PKA) activation, then Rap-1 and B-Raf, leading to ERK1/2 activation. For $G_{\alpha q/11}$-coupled receptors, they typically use a protein kinase C (PKC)-dependent mechanism to directly activate Raf1. Many $G_{\alpha q/11}$-coupled GPCRs, such as the AT1R, also couple to $G_{\alpha i}$ and may activate ERK1/2 through a Ras-dependent mechanism that involves the βγ subunit of the G protein and RTK transactivation[11–13]. Recently, it was shown that Gβγ from $G_{\alpha s}$-coupled receptors can activate ERK1/2 via Src/Shc/SOS/Ras pathway[14].

GPCR responses are also regulated by β-arrestins (β-arrestin 1 and 2, also known as arrestin 2 and 3, respectively), which not only restrict G protein-mediated signaling through a process of desensitization, but also act as endocytic adaptors to remove receptors from the PM by interacting with proteins of the clathrin-coated vesicles (CCVs, also referred to as clathrin-coated pits (CCPs) at the PM) like AP-2 and clathrin itself to promote GPCRs endocytosis[1,6,15]. Moreover, β-arrestins bind the small GTPase ARF6 which also aids in recruiting AP-2 and clathrin to regulate receptor internalization[16,17]. β-arrestins have also been shown to act as signaling adaptors by recruiting signaling proteins, notably elements of the MAPK cascade such as Raf-1, MEK, and ERK1/2 itself[18,19]. However, because β-arrestins themselves do not possess enzymatic activity, MAPK like Raf-1 must be activated by other mechanisms, such as by the activation G proteins or RTK transactivation[20]. Indeed, recent studies have reported the absence of ERK1/2 activation by GPCRs in cells lacking G protein activity, hence suggesting a role for G protein-mediated signaling in supporting MAPK signaling by β-arrestins (it has to be understood hereafter that reference to β-arrestin-dependent MAPK signaling does not imply the lack of involvement of G protein signaling in such process)[14,21,22]. Like for many GPCRs, AT1R signals at the PM through $G_{\alpha q/11}$ to activate MAPK[9,12,13,23], but many studies[19,24,25], including a recent one from our lab[23], have shown that internalized AT1R-β-arrestin complexes also contribute to sustained ERK1/2 activation. B2R, β2AR, and V2R also engage in MAPKs signaling via G protein activation, and β-arrestins, via their signaling–scaffold functions, have also been involved in continued ERK1/2 signaling for these receptors[26–29].

Since receptor internalization regulates signaling at the PM, a tight endosomal regulation of internalized receptors is also necessary for their recycling and the re-establishment of signaling at the PM[1,2]. Signaling and trafficking processes are often intrinsically intertwined[2]. For instance, ERK1/2-mediated phosphorylation of β-arrestins has been shown to not only promote GPCR redistribution from the PM to inside the cell, but also to stabilize receptor-β-arrestin complexes in endosomes for many GPCRs, including AT1R, B2R, and V2R, with the resulting effect of delaying recycling and reducing receptor-mediated signaling at the PM[23,28,30,31]. The development of selective inhibitors of receptor internalization/trafficking and/or signaling would therefore allow us to better understand how each of these cell processes regulate the overall GPCR responsiveness, and to what extent they are intertwined. In that regard, we recently took advantage of structural information on β-arrestin interacting with AP-2, through its β-appendage, to computationally dock a library of small molecules and identified Barbadin, an inhibitor that retains receptors in CCPs and prevents MAPK activation by V2R[26]. To expand our pharmacological arsenal of inhibitors, we aimed at targeting specifically the endosomal receptor/β-arrestin complex and used the prototypical AT1R. However, with a lack of structural information available on the AT1R/β-arrestin complex, we resorted to unbiasedly screen libraries of small compounds using an endosomal GPCR trafficking assay, which we previously showed is amenable to high-throughput (HT) formats[32].

Here, we describe the serendipitous discovery of a dual Ras and ARF6 inhibitor, which we name Rasarfin. Our findings suggest that Rasarfin blocks GPCR internalization through inhibition of ARF6, a regulator of CCV-mediated endocytosis, and receptor-mediated MAPK through a novel Ras inhibition mechanism.

## Results

**Screening and characterization of inhibitors of GPCR internalization.** We performed a HT campaign in HEK293 cells using the bystander endosomal BRET sensor, which consists of a *Renilla* luciferase (RlucII)-tagged AT1R and an acceptor *renilla* GFP (rGFP) anchored to endosomes through an Endofin-FYVE (Endo-rGFP) domain[12]. AngII-mediated trafficking of AT1R to endosomes was screened against commercial libraries of ~115,000 small molecules (Fig. 1a and Supplementary Data 1). Because our assay is based on luminescence and fluorescence, we excluded 6610 compounds that quenched more than 50% of the luciferase or GFP signals. After excluding 5 AT1R antagonists[12], we conserved hit compounds blocking by more than 40%, as well as those potentiating by more than 100% receptor internalization. From this collection of 943 compounds, a second validation on AT1R internalization was performed using the same BRET assay, as well as on the agonist-mediated B2R internalization, this time using B2R-RlucII and Endo-rGFP. Only compounds that were revalidated on AT1R and affected B2R endocytosis were kept, which included forty hits (20 potentiators and 20 inhibitors) (Fig. 1b and Supplementary Data 2). We focused on inhibitors that were commercially available at the beginning of the project and tested their cell toxicity by microscopy, looking at cells expressing B2R tagged with YFP, which highlights cell contours and potential morphological changes, such as rounding up of cells following treatment with 50 μM of the compounds that may have indicated cell death. None of the inhibitors tested caused abnormal cell morphological changes when cells were treated with compounds as compared to DMSO-treated cells. Inhibitors were also assessed in terms of structure novelty, derivatization potential, and commercial availability of analogs (within 95–98% similarity from Tanimoto metric, Sci-Finder, https://scifinder-n.cas.org), for structure-activity relationship studies (SAR). They were scored 1–5, from best to worst. We excluded compounds with potential nucleophilic reactivity (e.g., Michael addition:

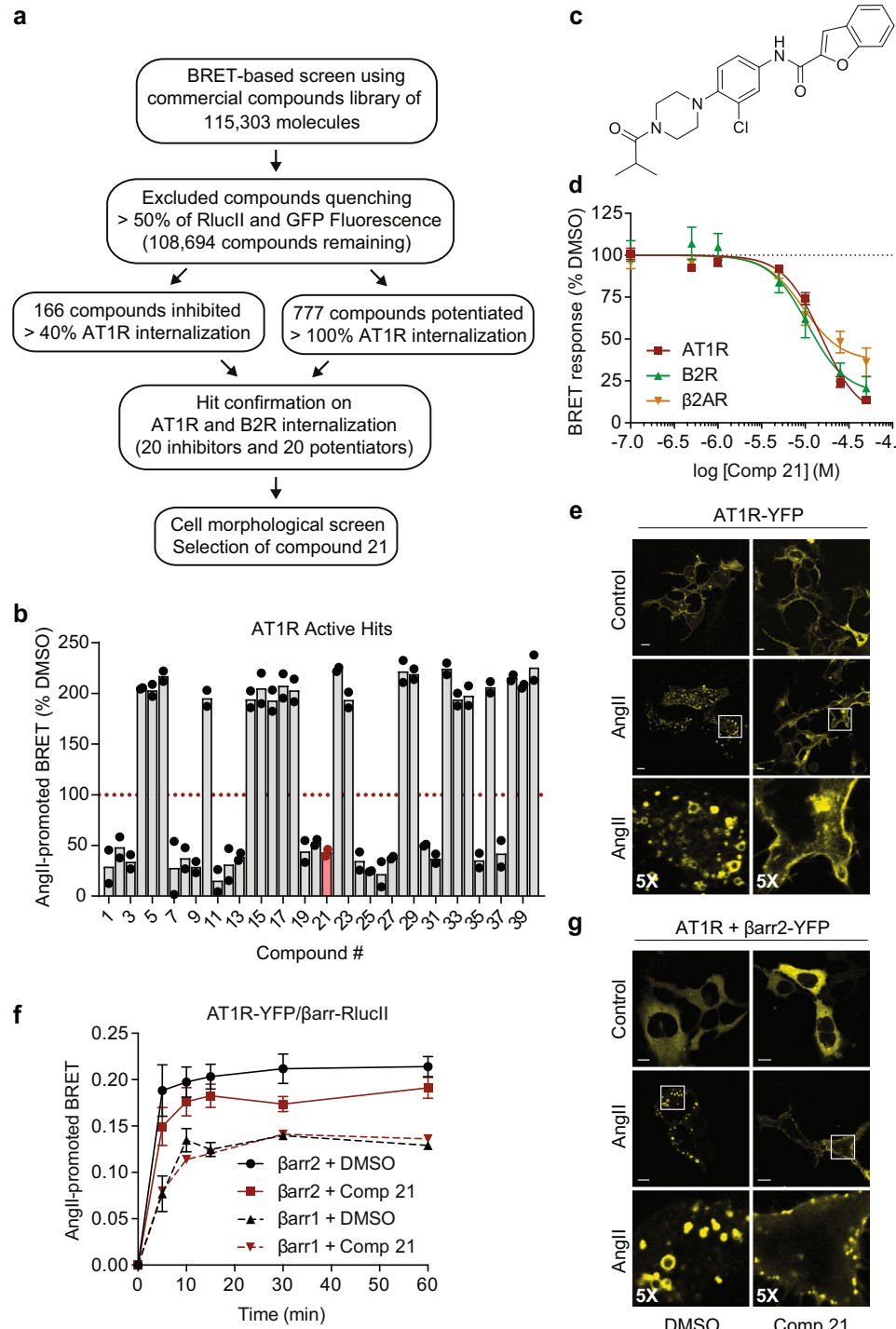

**Fig. 1 High-throughput screening identifies compound 21 as an inhibitor of GPCR internalization. a** Flowchart outlining the steps of the screening and selection of hits to the lead compound. **b** HTS results of 40 active hits on AT1R internalization using trafficking sensors. BRET responses were quantified as percent AngII-promoted BRET compared to DMSO and are presented as individual values, $n = 2$ biologically independent experiments. **c** Structure of the selected compound **21**. **d** Effects of **21** (50 μM) on AT1R (closed red squares and line), B2R (closed green triangles and line), and β₂AR (closed orange triangles and line) internalization into endosomes. BRET responses were quantified as percent ligand-promoted BRET compared to DMSO. The mean values of the ligand-promoted BRET responses ($BRET_{ligand}$–$BRET_{basal}$) in DMSO were 0.260, 0.549, and 1.061 for AT1R, B2R and β₂AR, respectively. Data are presented as the means values ± SEM, $n = 4$ biologically independent experiments performed in triplicate. **e** Confocal microscopy images of YFP-tagged ATIR internalization, repeated independently three times with similar results. Scale bar = 10 μm. Bottom micrographs are 5× enlargements of the boxed areas. **f** BRET recording of the recruitment of β-arrestin1 and β-arrestin2 to AT1R in the absence (DMSO, black triangles and dotted line and black circles and solid line, respectively) or presence of **21** (50 μM, red triangles and squares, respectively, and lines). BRET responses were quantified as AngII-promoted BRET. Data are presented as mean values ± SEM, at least $n = 3$ biologically independent experiments performed in triplicate. Source data are provided as a Source Data file. **g** Confocal microscopy images of YFP-tagged β-arrestin2 internalization, repeated independently three times with similar results. Scale bar = 10 μm. Bottom micrographs are 5× enlargements of the boxed areas.

compounds **7**, **26**, and **27**), those reported to have biological activities (compounds **8**, **20**, **25**, **30**, and **35**) and/or showing autofluorescence (compound **26**). Compound **19** ranked high, but only had one other molecule with a similar structure available. Although other molecules could be interesting in their own rights as endocytosis inhibitors (e.g., compounds **1**, **9**, **11**, **12**, **13**, **24**, **31**, and **37**), we focused on compound **21** (Fig. 1c) because not only did it rank high in our analysis, but it was also the only compound with substantially more similar analogs commercially available at the time of these investigations. In the screen and the follow-up validation assay, **21** inhibited more than 50% AT1R internalization and more than 65% B2R internalization (Supplementary Data 2). $IC_{50}$s for the receptor internalization inhibition of AT1R and B2R were respectively 10 and 11 µM (Fig. 1d); and as revealed by microscopy, these receptors remained trapped at the PM when cells were incubated with **21** (Fig. 1e and Supplementary Fig. 1a). **21** also inhibited isoproterenol (Iso)-mediated β2AR internalization into endosomes, with an $IC_{50}$ of 15 µM (Fig. 1d), suggesting a common inhibitory mechanism, albeit independent of the compound's direct action on these receptors. Because β-arrestins are critically involved in GPCR endocytosis and trafficking in cells, we assessed **21**'s effects on β-arrestins' recruitment to AT1R and B2R at the PM using a BRET assay[32], as well as their trafficking into endosomes by microscopy. **21** did not preclude β-arrestin1 or β-arrestin2 recruitment to AT1R or B2R at the PM (Fig. 1f and Supplementary Fig. 1b) but prevented β-arrestins' trafficking with receptors into endosomes (Fig. 1g and Supplementary Fig. 1c). β-arrestin2-YFP accumulated in endosomes of AngII- and BK-stimulated cells in absence of compound **21** treatment but remained in punctuated structures at the PM, reminiscent of CCPs[26,33], when the inhibitor was present.

### Compound 21 targets ARF6 to inhibit receptor internalization.
Compound **21**'s effects on β-arrestin's ability to recruit AP-2 was assessed since our findings suggested that it inhibited processes occurring between β-arrestin's binding to receptors at the PM and CCV-dependent internalization of the complex. HEK293 cells were transfected with a BRET sensor that monitors the interaction between β-arrestins and the β-appendage of AP-2 (Fig. 2a)[26,33]. We compared the effect of Barbadin, at its maximal solubility concentration of 100 µM, which we show inhibited more than 50% of the formation of the β-arrestin/AP-2 complex mediated by AT1R (Fig. 2a and ref. [26]) and blocked receptor internalization, to that of **21**'s effect on such complex formation. Barbadin was non-toxic to cells since HEK293 treated cells retained their flat, cobblestone appearance[26]. **21**, at 50 µM, also blocked around 90% AT1R internalization (Fig. 1d) but only inhibited no more than 20% of the AngII-mediated β-arrestin/AP-2 complex formation. The small GTPase ARF6, which binds β-arrestins and facilitates AP-2 and clathrin recruitment at the PM, is also necessary for CCPs initiation and receptor internalization[17]. Therefore, ARF6 activity was assessed using the protein-3 binding domain (PBD) of the golgi-associated, gamma adaptin ear-containing, ARF-binding protein 3 (GGA3) fused to GST (GST-GGA3-PBD) in a pull-down assay, which allows the detection of GTP-bound ARF6 in cells[34] (Fig. 2b). Quantification of ARF6 activity revealed an increase of fourfold in ARF6-GTP in cells stimulated with AngII and more than 70% inhibition with **21**. This inhibition was selective for ARF6 since **21** had no effect on AT1R-dependent ARF1 activation (Supplementary Fig. 2a). We also devised a BRET assay to quantitatively assess **21**'s action on ARF activity. The ARF BRET assay was created using the GGA3-PBD fused to RlucII (GGA3-PBD-RlucII) and the rGFP anchored at the PM (rGFP-CAAX) (Supplementary Fig. 3a). AngII stimulation of HEK293 cells expressing AT1R and the ARF

BRET sensor produced a maximum signal that was not further enhanced by the overexpression of ARF6, but was readily inhibited by the expression of the dominant-negative (DN) ARF6-T27N (Supplementary Fig. 3b). **21** blocked ARF's activation by 40–60% following AT1R stimulation (Fig. 2c). Overexpressing ARF6 and ARF6-T27N respectively increased and decreased the kinetics and extent to which agonist-bound AT1R was removed from the PM (Fig. 2d). **21** had similar effects as ARF6-T27N in that they both slowed down and decreased the extent of AngII-bound AT1R removal from the PM (Fig. 2e), suggesting **21** targeted this small G protein to inhibit CCP-mediated internalization of receptors.

### Compound 21 blocks GPCR- and RTK-mediated signaling.
We reasoned that because internalizing AT1R and its targeting to intracellular compartments can support continued MAPK signaling, and since **21** inhibits receptor internalization, it should also impede to some extent ERK1/2 activation. Surprisingly, **21** not only totally inhibited AngII-mediated phosphorylation of ERK1/2 but it also blocked MAPK responses promoted by other receptors, like the B2R, β2AR, and EGFR (Fig. 3a). ERK1/2 activation by AT1R and EGFR were inhibited with similar potencies ($IC_{50}$ of 5 µM and 4 µM, respectively) (Fig. 3b, c), also suggesting a common mechanism of signaling inhibition across classes of membrane receptors. **21** efficiently inhibited ERK1/2 activation by AT1R in cells lacking $G_{αq/11}$ and either supplemented or not with β-arrestin, which favor more receptor internalization, hence potentially limiting G protein-dependent signaling at the PM and favoring more β-arrestin–scaffold MAPK signaling. It also similarly inhibited ERK1/2 activation in β-arrestin1/2 KO cells, where AT1R signaling occurs at the PM because of lack of receptor internalization and desensitization (Supplementary Fig. 4a). Together, these results suggest that **21** acts through a common effector of these pathways to inhibit AT1R-mediated ERK1/2 activation in these KO cells. The inhibitory action of **21** on ERK1/2 activation was not due to the direct inhibition of $G_{αq}$, $G_{αi3}$, or $G_{α12/13}$ engagement by AT1R, as revealed by the lack of inhibitory effects on BRET sensors, which report on either these G proteins activity or their downstream effectors[12](Supplementary Fig. 4b). Receptor-mediated ERK1/2 signaling can involve upstream kinases such as PKC, BRAF, and MEK1[35,36]. Consistent with the lack of effect of **21** on $G_{αq}$, it did not inhibit AT1R-mediated PKC activation (Supplementary Fig. 4b), nor did it block ERK1/2 activation induced by the overexpression of either the active forms of MEK1 (MEK1-DD) or BRAF (BRAF-V600E) (Supplementary Fig. 4c, d). **21** also inhibited EGFR-mediated phosphoinositide 3-kinases (PI3K)-Akt signaling, as assessed by western blot (Fig. 3b, d) or with a BRET assay[37], using the PH domain of Akt (Akt (PH)-RlucII) and the rGFP-CAAX (Supplementary Fig. 5). In both assays, **21** inhibited Akt with similar potencies (4–5 µM), which mirrored ERK1/2 inhibition (~5 µM), suggesting again a common target involved in both signaling pathways. To exclude other kinases potentially involved in ERK1/2 and Akt activation as targets of **21**, we ran a kinase binding assay (KINOMEscan™) on 384 kinases from the human genome[38]. Remarkably, a very high selectivity score S(10) of 0.008 was found (Supplementary Data 3). Notably, **21** failed to bind EGFR, BRAF, different Akt, PI3K, MEK, and ERK subtypes, nor did it interact with G protein-receptor kinases (GRKs 1–4 and 7), consistent with its lack of inhibitory effects on β-arrestin recruitment to agonist-occupied receptors at the PM. **21** also failed to bind to the non-receptor tyrosine kinase Src and the adaptor-associated protein kinase 1 (AAK1), both regulators of CCV-mediated internalization[39,40].

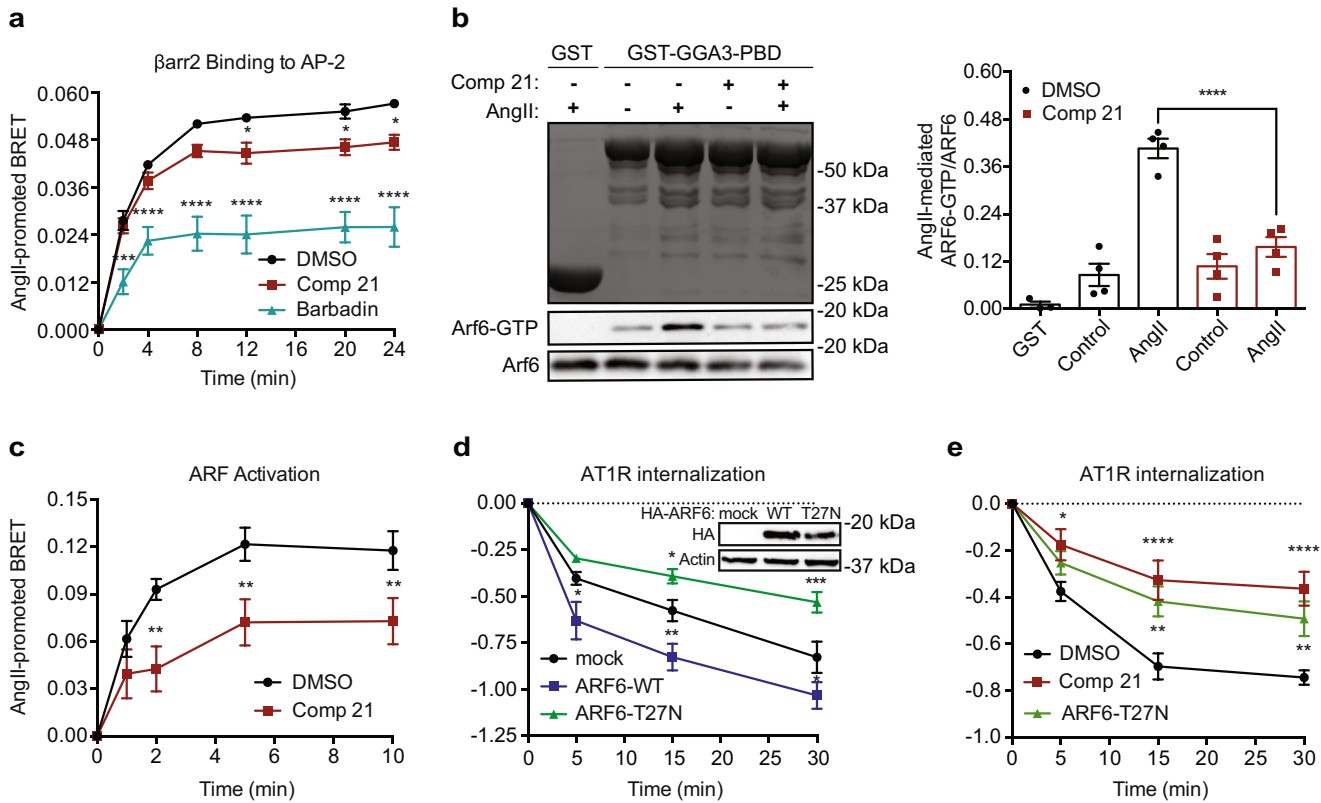

**Fig. 2 Compound 21 targets ARF6 to inhibit GPCR internalization. a** Effects of **21** (50 µM, closed red squares and line) and Barbadin, a β-arrestin2/AP-2 inhibitor (100 µM; closed cyan triangles and line), on the binding of β-arrestin2 to AP-2. BRET responses were quantified as AngII-promoted BRET and are presented as mean values ± SEM, $n = 3$ biologically independent experiments performed in triplicate, $*p < 0.05$, $***p < 0.005$, $****p < 0.0001$, two-way ANOVA corrected with Dunnett's test. **b** Coomassie of GST and GST-GGA3-PBD proteins and western blots of AT1R-mediated HA-ARF6 activation as assessed by glutathione beads pull-downs. Right panel is the quantification of AT1R-mediated ARF6 activation in the absence (DMSO, open black bars) or presence of **21** (50 µM, open red bars) calculated as the amount of ARF6-GTP over total ARF6 and are presented as the mean values ± SEM, $n = 4$ biologically independent experiments, $****p < 0.0001$, one-way ANOVA with Bonferroni correction. **c** BRET kinetics of AT1R-mediated ARF activation in the absence (DMSO, closed black circles and line) or presence of **21** (50 µM, closed red squares and line). Data were quantified as AngII-promoted BRET and are represented as the mean values ± SEM, at least $n = 3$ biologically independent experiments performed in triplicate, $**p < 0.01$, two-tailed unpaired Student's t-test. **d** BRET recordings of AT1R internalization represented as the removal of AT1R from the PM. Cells were transfected with an empty vector (mock, closed black circles and line), HA-ARF6-WT (closed blue squares and line), or HA-ARF6-T27N (closed green triangles and line). Data were quantified as AngII-promoted BRET and are presented as the mean values ± SEM, at least $n = 3$ biologically experiments performed in triplicate, $*p < 0.05$, $**p < 0.01$, $***p < 0.005$, two-way ANOVA corrected with Dunnett's test. Western blots of HA and β-actin in the inset are used as controls of protein expression. **e** BRET recordings of AT1R removal from the PM in cells transfected with ARF6-T27N (closed green triangles and line) or empty vector (closed black circles and line) and incubated or not with **21** (50 µM, closed red squares and line). Data were quantified as AngII-promoted BRET and are presented as the mean values ± SEM, at least $n = 3$ biologically independent experiments performed in triplicate, $*p < 0.05$, $**p < 0.01$, $****p < 0.0001$, two-way ANOVA corrected with Dunnett's test. Source data are provided as a Source Data file.

**Compound 21 inhibits Ras.** Since **21** inhibited both ERK1/2 and Akt activation by GPCRs and EGFR, we looked at a common target involved in both pathways. The small G protein Ras is an upstream regulator of both Ras-RAF-MEK-ERK and PI3K/Akt signaling pathways[35,36]. Therefore, we explored **21**'s ability to inhibit Ras activity using GST-Raf1-Ras-binding domain (RBD) in a pull-down assay[41]. A reduction of 60% in GTP-bound Ras was noticed in pulled-down from cells treated with **21** and stimulated with either AT1R or EGFR, as compared to vehicle-treated cells (Fig. 4a, b). Complementary to this in vitro assay, we generated a Ras BRET assay, using the Raf-Ras binding domain tagged with RlucII (Raf1-RBD-RlucII) and the rGFP at the PM (rGFP-CAAX) (Supplementary Fig. 3c), to assess **21**'s effects on Ras activation kinetics. Following AT1R stimulation, the Ras sensor generated a quantifiable BRET signal that was efficiently inhibited by the overexpression of the DN Ras-S17N. Overexpression of either the active form of Ras (G12V) or SOS1 anchored at the PM (SOS-CAAX) produced a robust basal BRET

signal that was no longer modulated by receptor activation (Supplementary Fig. 3d). Using this Ras BRET assay, we found that **21** decreased the small G protein activation by 50–70% following AT1R stimulation (Fig. 4c). In a dose-response experiment using the ARF and Ras BRET sensors, **21** respectively inhibited these small G proteins with IC$_{50}$s of 7 and 0.7 µM (Fig. 4d). **21**'s potency to inhibit ARF6 and Ras is also consistent with its relative efficacy for inhibiting AT1R internalization and MAPK, respectively, when used at 50 µM (e.g., around 7 and 70 times the respective IC$_{50}$s) (Figs. 2e and 3a). We also assessed if **21** was acting as a pan small G protein inhibitor by testing its effect on Rho and Rac/Cdc42. Up to 100 µM of **21** did not inhibit AT1R-mediated activation of a Rho BRET sensor[12], nor did it prevent the pull-down of Rho-GTP by GST-Rhotekin-RBD from cells stimulated with AT1R (Fig. 4d and Supplementary Fig. 2b). To assess Rac/Cdc42 activity, we generated a Rac/Cdc42 BRET sensor that consisted of the PAK-CRIB domain fused to RlucII (PAK-CRIB-RlucII) and the rGFP-CAAX (Supplementary

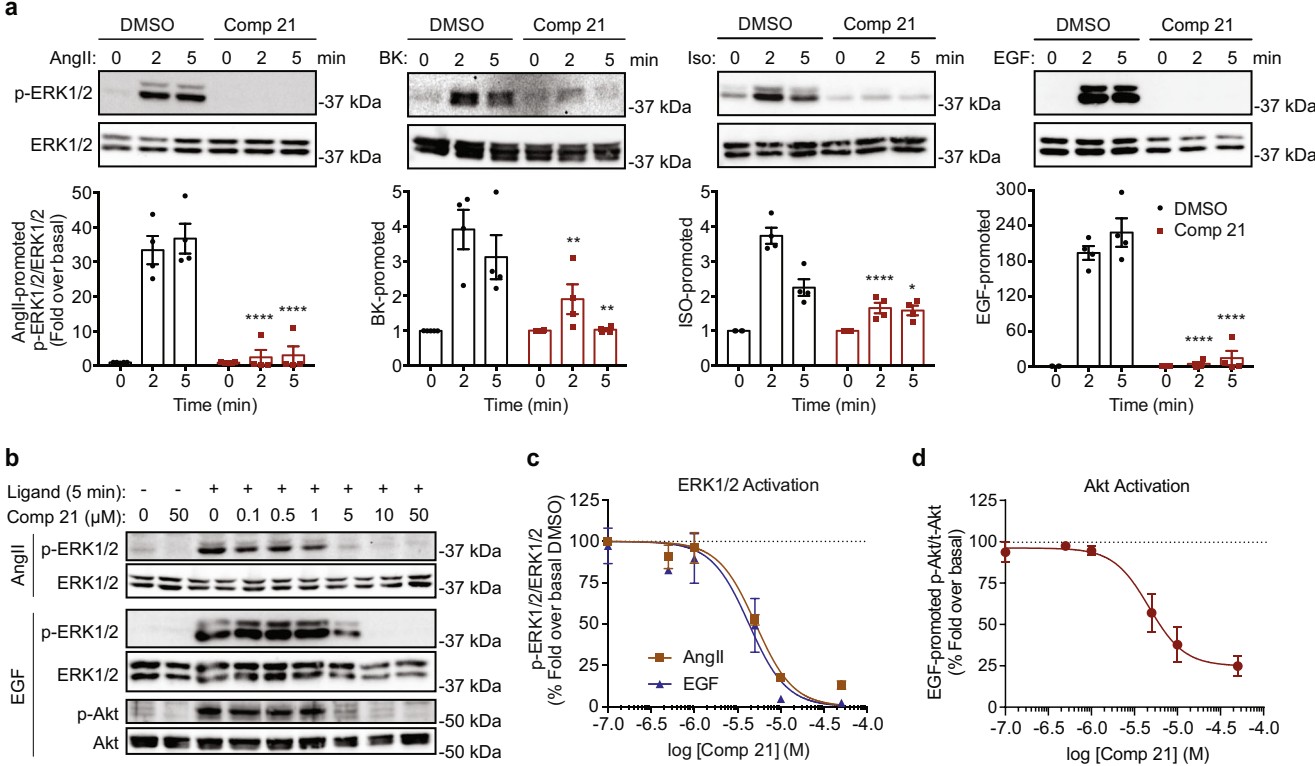

**Fig. 3 Compound 21 inhibits receptor signaling. a** Western blots of ERK1/2 phosphorylation kinetics in cells mediated by AT1R, B2R, β₂AR, and EGFR in absence (DMSO, black open bars) or presence of **21** (50 μM, red open bars). Data were quantified as p-ERK1/2 over ERK1/2, normalized to basal (0 min), and compared to DMSO. Data are presented as mean values ± SEM, $n = 4$ biologically independent experiments, $*p < 0.05$, $**p < 0.01$, $****p < 0.0001$, two-way ANOVA with Bonferroni correction. **b**–**d** Western blots and quantification of AT1R- and EGFR-mediated ERK1/2, and EGFR-mediated Akt activation in the absence (DMSO) or presence of **21** (at indicated concentrations). Data were quantified as AngII- (closed red squares and line) and EGF-mediated (closed blue triangles and line) **c** p-ERK1/2 over ERK1/2, and EGF-mediated (closed red circles and line) **d** p-Akt over Akt, respectively, and normalized as fold over basal, percent compared to DMSO (dotted lines). Data are presented as mean values ± SEM, $n = 3$ biologically independent experiments. Source data are provided as a Source Data file.

Fig. 3e). PAK-CRIB, which binds to active Rac/Cdc42, was recruited to the PM upon AngII stimulation of cells and generated a robust BRET signal, which was blocked by the overexpression of the DN Rac1-T17N. Overexpression of the constitutively active Rac1-Q61L increased the basal BRET, which was no longer modulated by AT1R stimulation (Supplementary Fig. 3f). Analogously to what we observed with Rho, **21** failed to inhibit Rac/Cdc42 activation mediated by AT1R (Fig. 4d). Since **21** selectively inhibits the two small G proteins Ras and ARF6, we renamed it Rasarfin (**Ras**-**ARF**-**in**hibitor).

Rasarfin's direct action on Ras was also validated using purified H-Ras in an in vitro GEF exchange assay using fluorescent mant-GTP[42]. Rasarfin inhibited in a dose-dependent manner the uptake of GTP into Ras in presence of its GEF, SOS1 (Fig. 4e). It acted directly on Ras since it also inhibited mant-GTP binding promoted by EDTA, which facilitates GDP-GTP exchange from Ras by chelating Mg²⁺ ions, independently of its GEF[43] (Fig. 4f). In these conditions, Rasarfin inhibited Ras GTP loading, although with seemingly less potency and overall efficacy than when SOS was used to promote the nucleotide exchange on Ras.

**Rasarfin targets Ras to inhibit signaling and ARF6 to block receptor internalization.** Because Rasarfin inhibited both receptor internalization and signaling, the relative contribution of Ras and ARF6 in both processes was next investigated. First, the agonist-mediated removal of AT1R from the PM was assessed in cells expressing wild-type K-Ras, another isoform that is highly homologous to H-Ras[44,45], or K-Ras-S17N along with the

receptor. No differences in the kinetics or extents of AngII-mediated AT1R internalization were found between these conditions and as compared to mock-transfected cells (Supplementary Fig. 6a), suggesting that Ras does not regulate AT1R endocytosis, like ARF6 does (Fig. 2d). Next, cells were transfected with either ARF6-T27N or K-Ras-S17N along with AT1R, and MAPK activation was measured following receptor simulation. ARF6-T27N did not have a significant effect on AngII-mediated ERK1/2 stimulation, while K-Ras-S17N efficiently inhibited the mitogenic response (Supplementary Fig. 6b, c).

**Molecular dynamics simulations support the binding of Rasarfin within the SOS binding domain of Ras.** We took advantage of the many Ras structures available to gain insight on Rasarfin's binding to this small G protein using molecular dynamics (MD) simulation. We used the SOS1-bound X-ray structure of H-Ras (PDB: 1BKD) since our findings suggested that Rasarfin interfered with the ability of SOS to promote guanine nucleotide exchange on Ras. Rasarfin was, therefore, docked into the binding groove between switch I and switch II on Ras (residues 25–40 and 57–75, respectively), where SOS normally binds. The eight best-scored docking poses suggested a very similar orientation of the ligand inside the binding groove (Fig. 5a and Supplementary Fig. 7a–h), with only 2 out of the 10 best poses showing inverted binding (Supplementary Fig. 7i, j). Clustering the poses resulted in 2 clusters (Fig. 5a and Supplementary Fig. 7k) and analysis of the poses resulted in a best interaction profile, which displayed the chloro atom turned inside

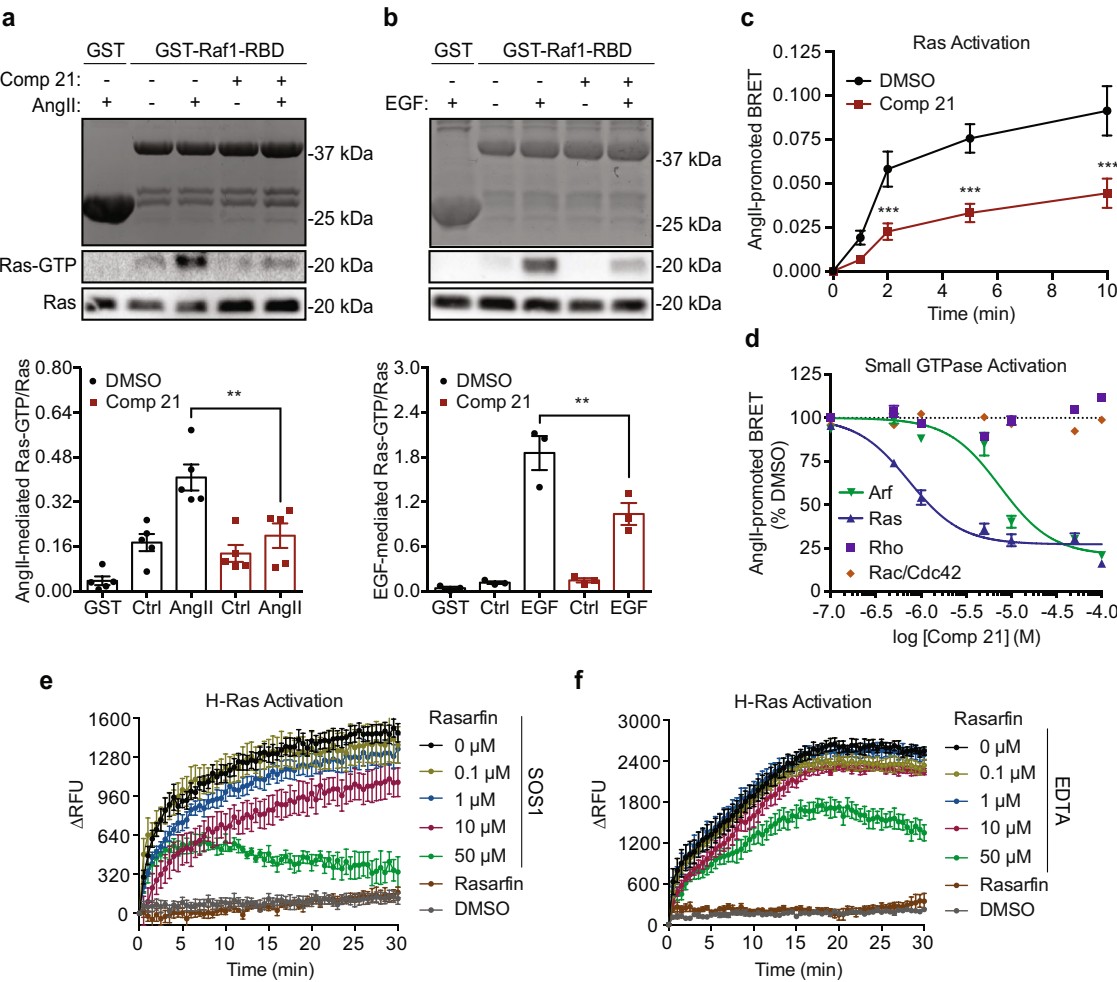

**Fig. 4 Compound 21 targets Ras. a** Coomassie of GST and GST-Raf1-RBD proteins and western blots of AT1R-mediated and **b** EGFR-mediated Ras activation in absence (DMSO, open black bars) or presence of **21** (50 μM, open red bars) as assessed by GST- and GST-Raf1-RBD-coupled to glutathione beads pull-downs. Ras activation was calculated as the amount of Ras-GTP over the total Ras detected and are presented as mean values ± SEM, $n = 5$ for AT1R (**a**) and $n = 3$ for EGFR (**b**) biologically independent experiments, $**p < 0.001$, one-way ANOVA with Bonferroni correction. **c** BRET kinetics of AT1R-mediated Ras activation in absence (DMSO, closed black circles and line) or presence of **21** (50 μM, closed red squares and line). Data were quantified as AngII-promoted BRET and are presented as mean values ± SEM, $n = 4$ biologically independent experiments performed in triplicate, $***p < 0.005$, two-tailed unpaired Student's $t$-test. **d** BRET recording of the activation of Rho (purple squares), Rac (orange diamonds), ARF (green triangles and line) and Ras (blue triangles and line) by AT1R in the presence of **21** (50 μM). BRET responses were quantified as AngII-promoted BRET, percent compared to DMSO (dotted line) and are presented as mean values ± SEM, $n = 3$ (Rho and ARF) and $n = 4$ (Rac and Ras) biologically independent experiments performed in triplicate. **e**, **f** In vitro kinetics of mant-GTP loading onto H-Ras. Purified H-Ras nucleotide exchange was induced using **e** purified SOS1 or **f** 40 mM EDTA in the presence of DMSO (black dots and lines) or different concentrations of Rasarfin (as indicated with respective colors). The relative fluorescence unit (RFU) was measured every 30 s for 30 min and quantified as the delta RFU (RFU post-addition minus the 5 averaged RFU pre-addition, per condition). Data are presented as mean values ± SEM, at least $n = 4$ biologically independent experiments. Source data are provided as a Source Data file.

the Ras binding pocket. To support the position of the ligand inside the groove, we also simulated the spontaneous association of Rasarfin to Ras. Five Rasarfin molecules were randomly placed in the solvent around the open Ras (Fig. 5b) and simulated for a total of 6 μs (3 × 2 μs), yielding an average volumetric map of ligand positions (Fig. 5c). This map reveals that during spontaneous association, Rasarfin favors the identified Ras groove as its preferred binding site. To assess further the stability of Rasarfin binding in the interface of Ras between switch I and II, we extended the simulation to a total time of 12 μs (3 × 4 μs). Monitoring the Root Mean Square Deviation (RMSD) of Rasarfin shows that Rasarfin can stably bind to the predicted Ras binding groove for microseconds (Fig. 6a, replica 1 for 1.5 μs and the whole 4 μs of simulation in replica 2) but also unbinds at times (Fig. 6a, end of replica 1 and replica 3). The computation of contact frequencies from simulation data highlights key

pharmacophore interactions (Fig. 7a, b) that stabilize Rasarfin binding in the Ras' effector binding interface between switch I and II (Supplementary Video 1). The most discriminative features included the hydrophobic interactions (hereafter referred to as "H" groups) of the benzofuran (H1), the aryl (H2), the chloro-atom (H3), the isopropyl moiety (H4), and the H-bond donor (HBD) or H-bond acceptor (HBA) of the amide, which are the most detected combinations in the simulation. The benzofuran was most of the time stabilized by the hydrophobic interaction (H1) with Tyr40 or Leu56. The carbonyl oxygen of the backbone of Ile55 is an acceptor for the HBD of Rasarfin. The aryl ring with the attached chloro-atom (H2) excessively interacts with Thr20, Tyr40, and Ile55 (Fig. 7b, c), while H3 (the chloro-atom) is kept in place and in close vicinity to Val8, Thr20, Ile24, Tyr40 and Ile55, most of the simulation time. The isopropyl (H4) was mainly oriented toward switch II, in proximity to Ile21 (Fig. 7c).

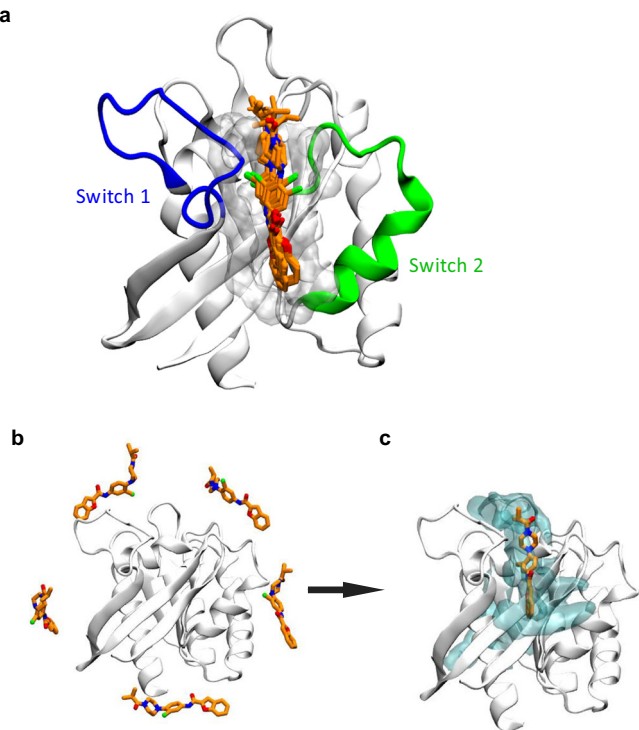

**Fig. 5 In silico studies reveal interactions between Rasarfin and Ras.**
**a** Shown are the 8 best-scored docking poses of Rasarfin in licorice representation (orange) bound in the groove between switch I and switch II on Ras (protein in white cartoon, binding groove surface in silver). **b–c** Unbiased simulations of Rasarfin association to Ras. **b** Shown is the initial placement of Rasarfin molecules (orange licorice) around Ras to study the spontaneous association of the ligand to Ras (white depiction). **c** Spontaneous association of Rasarfin to Ras approximated using a volumetric map of average ligand occupancy at 0.15 threshold (cyan transparent surface). The binding mode of Rasarfin obtained through docking is overlayed for comparison (orange licorice).

Superimposition of Rasarfin onto SOS1-bound Ras revealed that Rasarfin is covering a large part of the protein–protein interface (Fig. 6c) of Ras and SOS. The alkyl chain of Lys939 of SOS reaches into the binding groove of Ras, and its primary amine functions as a H-bond donor interacting with the backbone carbonyl of Asp57 in Ras (Fig. 6c). Rasarfin displays a similar orientation inside the binding groove like the elongated, upwards oriented Lys939. The frequency interaction of the carbonyl of the amide on Rasarfin with the backbone of Asp57 is high (e.g., 59%) (HBA1, Fig. 7a, c). The furan moiety of the benzofuran is overlapping with His911 in SOS, suggesting that Rasarfin can interact similarly with Ras like the residue on SOS. Tyr40 is oriented in a parallel fashion to His911, as well as to the benzofuran (Fig. 6c).

**The Cl-substitution in Rasarfin maintains its binding and inhibitory activity.** Detailed analysis of the proposed binding mode of Rasarfin reveals that the Cl-substitution attached to the aryl ring occupies a small cavity within the Ras protein (Fig. 6b), where it forms multiple interactions (H3, Fig. 7c). Removal of this atom leads to a binding mode where the binding groove between switch I and II on Ras remains occupied by the small molecule (Fig. 6b), but the affinity towards Ras is expected to be reduced. This is indeed supported by experimental data with compound **21.4**, a close analog of Rasarfin that was present in our screen but did not inhibit AT1R internalization (Fig. 8a and Supplementary Fig. 8). When placed into the binding groove of Rasarfin, **21.4**

cannot occupy the described structural cavity which is occupied by the Cl-atom in Rasarfin (Fig. 6b), suggesting that **21.4** lacks Ras and MAPK inhibitory activities due to the absence of the Cl-atom (Fig. 8b, c). Furthermore, MD simulation of the **21.4**-Ras complex reveals decreased stability of the compound binding compared to Rasarfin as indicated by overall high ligand RMSD values for replicates 1–3 (Fig. 6a and Supplementary Video 2). This is exemplified by **21.4** interactions with Ras, which exhibit patterns of only 2-feature pharmacophores binding (i.e., H1 and H2, Supplementary Fig. 9a, b). The bulkiness of the Cl-substituent on Rasarfin offers a conformational restraint, which leads to the Chloro-phenyl-piperazine moiety being in a locked conformation and properly positioning the isopropyl group. Furthermore, the aromatic ring (H2) of **21.4**, which is not restricted by the bulky substituent, was able to turn, allowing the piperazine to rearrange and significantly displacing **21.4** from the binding groove (Supplementary Video 2). We next evaluated the impact of Rasarfin binding on the dynamics of Ras switch I and switch II (Supplementary Fig. 10). Interestingly, when Rasarfin binds between the two switch domains, those are locked in an open-like conformation, as compared to GTP-bound Ras (Supplementary Fig. 10a versus b). Root Mean Square Fluctuation (RMSF) reveal that switch I moves significantly less (Supplementary Fig. 10c, d), and both switches cannot close onto the binding groove, as Rasarfin intercalates between the two loops, blocking their interaction. This could also prevent nucleotide binding onto Ras and its activation.

Because Tyr32 and Tyr40, located on the switch I domain of Ras, are critical for stabilizing Rasarfin binding through interactions with both the chlorine and isopropyl moiety (Fig. 7a, c), we reasoned that substituting them for Ala residues would negatively impact Rasarfin's inhibitory action on Ras. These residues are also located far from the GTP binding site, hence potentially minimizing their effects on nucleotides binding to Ras. Because we wanted to assess the inhibitory action of Rasarfin over different Ras subtypes, we mutated the two tyrosine residues to an alanine in K-Ras (Y32A, Y40A, and Y32A/Y40A mutants) within the same Rasarfin binding site found in H-Ras ($Lys^5$-$Thr^{74}$). This domain and residues are perfectly conserved among the Ras family members[44,45]. WT K-Ras and K-Ras mutants were overexpressed along with AT1R in cells and AngII-mediated activation of Ras was assessed using the Ras BRET sensor (Supplementary Fig. 11a). All constructs expressed well, and despite an important reduction in AT1R-mediated activation observed with the double K-Ras-Y32A/Y40A mutant as compared to K-Ras and other mutants, only WT was significantly inhibited by Rasarfin. Furthermore, Rasarfin inhibited the SOS-mediated mant-GTP binding onto purified K-Ras, but not on K-Ras-Y32A (Supplementary Fig. 11b).

**SAR studies on Rasarfin identify functionally selective analogs.** To potentially improve the functional affinity and inhibitory selectivity of Rasarfin on Ras, we performed SAR studies with eight analogs of Rasarfin (compounds **21.1–21.8**, Supplementary Fig. 8). Expectedly, because of the lipophilic nature of the binding site on Ras, decreasing the compounds' overall hydrophobicity (logP order: **21.7** < **21.2** < Rasarfin < **21.8**), by modifying only the isopropyl and/or the benzofuran groups or removing the chloroatom on Rasarfin's aryl was detrimental to the compounds' inhibitory action on ERK1/2 activation by AT1R (Fig. 8b). Replacing the isopropyl by an ethyl group as in **21.2** or the benzofuran group with a thiophene as in **21.7** reduced hydrophobicity, compared to Rasarfin, and both compounds lost their ability to inhibit MAPK (Fig. 8b). Increasing the compound's hydrophobicity as in **21.8**, where the isopropyl was replaced by a n-propyl group, maintained its MAPK inhibitory property.

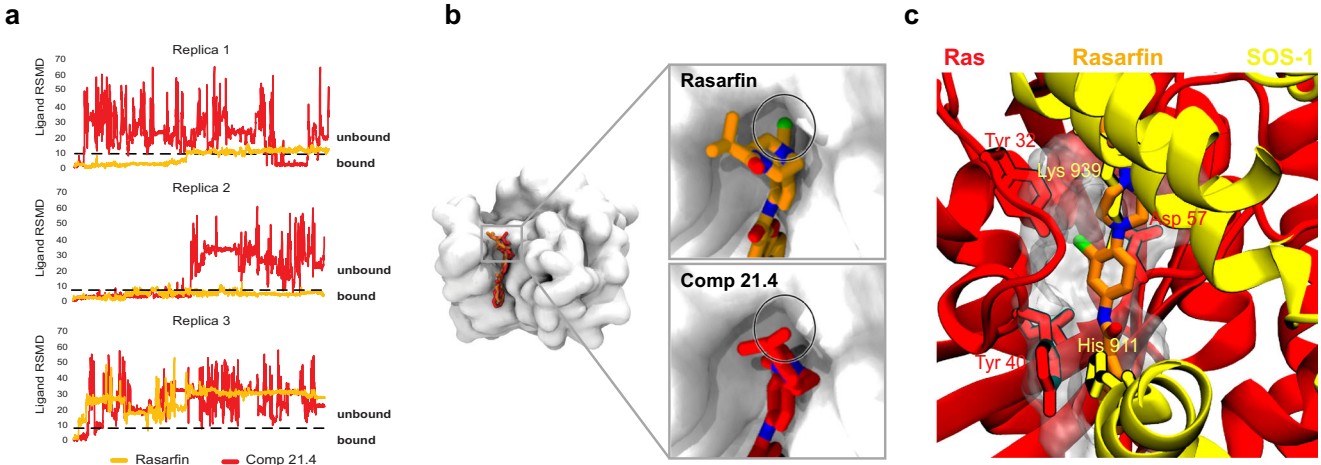

**Fig. 6 Estimation of the stability of Rasarfin and compound 21.4 binding modes. a** Root Mean Square Deviation (RMSD) (*y* axis) for Rasarfin (orange) and **21.4** (red) in respect to their initial position were analyzed in three separate MD runs. RMSD values higher than 6 correspond to simulation frames in which the ligand fully unbinds. The ligand position was monitored across 3 separate replicates of 4 μs (total 12 μs). **b** Comparison of predicted binding mode of Rasarfin (orange licorice) and compound **21.4** (red licorice). The molecular surface of Ras is represented in white and reveals the presence of a structural cavity in the vicinity of the predicted binding modes (black circle). In the predicted binding mode of Rasarfin, the structural cavity is occupied by the chlorine atom (in green), which is missing in compound **21.4**. **c** Overlay of Rasarfin binding into the Ras-SOS binding interface (PDB:1BKD). Rasarfin is displayed in orange licorice, SOS in yellow, and Ras in red ribbons. The surface occupied by Rasarfin is shown in transparent gray. Residues on Ras and SOS are highlighted in respective colors and labeled accordingly. Rasarfin is binding in the cavity which is normally occupied by residues His911 and Lys939 of SOS. Those residues are shown superimposed with Rasarfin, displaying His911 overlaying with the furan of the benzofuran moiety (AR1), and the Lysine elongated in the binding groove like Rasarfin, partially hidden as they overlay perfectly.

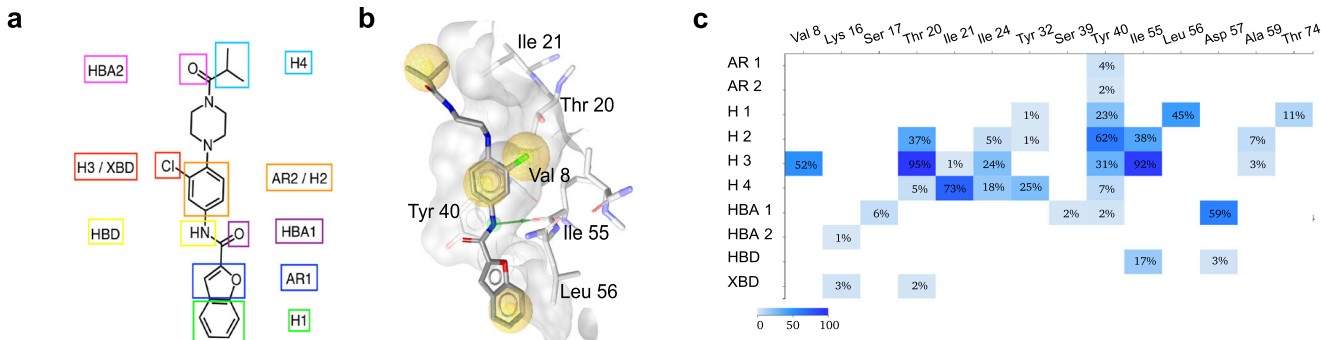

**Fig. 7 In silico assessment of Rasarfin interactions with Ras. a** 2D structure of Rasarfin labeled with established pharmacophoric features interacting with Ras residues. AR1-2 aromatic feature, H1-H4 hydrophobic feature, HBA1-2 hydrogen bond-acceptor, HBD hydrogen bond donor, XBD halogen bond donor. **b** Stick representation of Rasarfin embedded in the binding pocket of Ras (transparent gray) with interacting Ras residues labeled. Yellow spheres correspond to H1-H4; green arrow shows HBD of Rasarfin interacting with Ile55 backbone carbonyl. **c** Plot of interaction frequencies between residues of Ras (*x*-axis) and the pharmacophoric features of Rasarfin (*y*-axis). Color coding according to interaction frequency.

Consistent with the compound's effect on MAPK activity, **21.8** inhibited Ras exchange activity, while **21.4** and **21.7** had no effect (Fig. 8c). However, increasing or decreasing the overall hydrophobicity with **21.8** and **21.7**, respectively, both resulted in a loss of compounds' efficacy to inhibit AT1R internalization, as compared to Rasarfin (Fig. 8a). Not only are the physicochemical properties of the analogs coherent with Rasarfin's binding pose onto Ras, but this limited SAR analysis suggests that selective Ras inhibitors can be designed.

**Rasarfin inhibits Ras, ARF6, and cell proliferation in cancer cells.** Because Ras proteins are key regulators of normal cell proliferation and survival, inhibitors of this small G protein have been developed for the treatment of cancers[46–48]. We, therefore, assessed Rasarfin's effects on cancer cells viability and proliferation. We used MDA-MB-231 cells, a triple-negative breast cancer cell line derived from basal-like tumors with increased Ras, MAPK, and ARF6 activities[49–51]. MDA-MB-231 cells treatment

with Rasarfin resulted in a dose-dependent reduction in cell growth over time (Supplementary Fig. 12a). After 48 and 72 h of treatment, Rasarfin modestly decreased in a dose-dependent manner the metabolic activity of cancer cells compared to cells treated with DMSO. However, compared to Doxorubicin which potently killed cells, cell viability was not affected by Rasarfin treatment (Supplementary Fig. 12b). We also assessed to what extent Rasarfin inhibited proliferation of A549 lung cancer cells, which have increased Ras and EGFR activity[52]. Similar to the antiproliferative effect on MDA-MB-231 cells, we observed efficient A549 cell growth inhibition at 5 and 10 μM of Rasarfin, although we observed more important cell viability effects of Rasarfin on A549 cells than MDA-MB-231 cells at 10 μM of Rasarfin treatment (Supplementary Fig. 12c, d). We tested **21.8** on MDA-MB-231 cell growth and compared it to **21.4**. Despite both compounds moderately decreasing the metabolic activity of cancer cells, only **21.8**, which efficiently inhibits MAPK as well as Rasarfin, inhibited cell proliferation (Supplementary Fig. 12e, f). Rasarfin

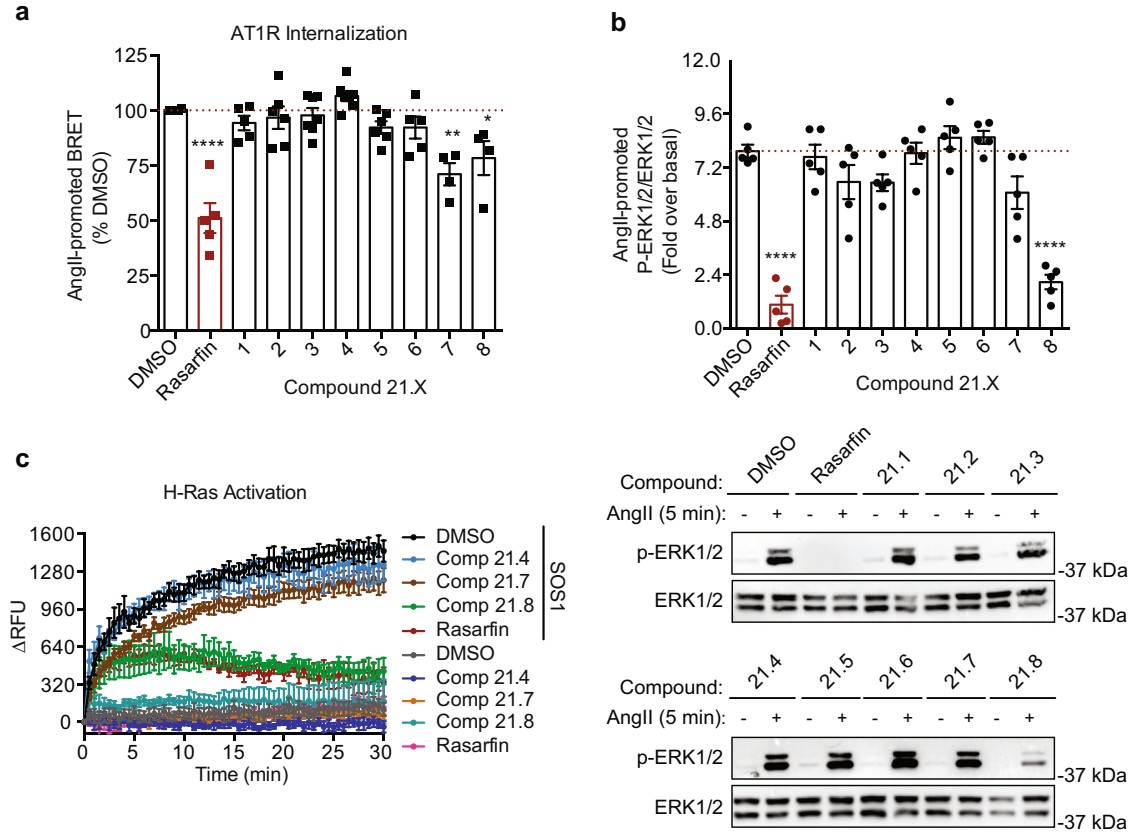

**Fig. 8 Functional selectivity of Rasarfin analogs. a** BRET recording of AT1R internalization into endosomes in the absence (DMSO, open black bar) or presence of 50 μM Rasarfin (open red bar) or compounds **21.1**–**21.8** (open black bars). BRET responses were quantified as AngII-promoted BRET response, percent compared to DMSO (red dotted line). Data are presented as mean values ± SEM, at least $n = 4$ biologically independent experiments performed in triplicate, $*p = 0.0234$, $**p = 0.0013$, $****p < 0.0001$, one-way ANOVA with Dunnett's test. **b** Quantification of AT1R-mediated ERK1/2 activation in cells the absence (DMSO, open black bar) or presence of 50 μM Rasarfin (open red bar) or compounds **21.1**–**21.8** (open black bars) and representative western blots. Data are quantified as p-ERK1/2 over ERK1/2 and normalized as fold over basal. Data are presented as mean values ± SEM, $n = 5$ biologically independent experiments, $****p < 0.0001$, one-way ANOVA with Dunnett's test. **c** In vitro kinetics of mant-GTP loading into H-Ras. Purified H-Ras was activated using purified SOS1 and in the presence of DMSO, Rasarfin, **21.4**, **21.7** or **21.8** (50 μM). The relative fluorescence unit (RFU) was measured every 30 s for 30 min and quantified as the delta RFU (RFU post-addition minus the 5 averaged RFU pre-addition, per condition). Data are presented as mean values ± SEM, $n = 3$ independent experiments. Source data are provided as a Source Data file.

also inhibited Ras and ARF6 activities in MDA-MB-231 cells (Supplementary Fig. 12g, h).

## Discussion

The present work has led to the unexpected identification of the first dual ARF6 and Ras small G protein inhibitor: Rasarfin. Our findings not only highlight the possibility of targeting these two small G proteins to inhibit GPCR internalization and receptor-mediated MAPK signaling but also suggest that new classes of inhibitors with distinct binding modalities can be developed to inhibit Ras and mitogenic responses in cells.

Extensive efforts have been devoted in developing Ras inhibitors for cancer therapies, with mixed results, which has led to the idea that Ras is perhaps "undruggable"[47,48,53]. Strategies included the prevention of nucleotide binding to Ras or its targeting to PM, the inhibition of Ras-effectors interaction, including its interaction with nucleotide exchange factors, and the stabilization of non-functional protein complexes with Ras. However, targeting protein–protein interfaces remains challenging due to the large contact surfaces of proteins and low binding affinity of molecules. Moreover, targeting the GTP binding site has also been difficult because of the high affinity of this nucleotide for Ras. We identified an inhibitor that interacts with Ras in a discrete manner to block SOS-mediated nucleotide exchange on Ras. With the recent

report of a β-arrestin/AP-2 inhibitor[26], the present study adds to the strategy aiming at developing new classes of inhibitors targeting protein–protein complexes for regulating receptor signaling and trafficking.

Previous studies have described inhibitors of the Ras-SOS complex that interact with Ras, albeit with relatively low affinities (e.g., mM range)[54–56]. These include the discovery of a peptide that mimicked a helical hairpin of SOS (helix H), which makes direct contact with Ras[55], and the use of NMR-based screens that identified small molecules that bind Ras between switch I and switch II domains[46,57,58]. The recently reported DCAI (4,6-dichloro-2-methyl-3-aminoethyl-indole) blocks SOS binding to Ras but only inhibits the nucleotide exchange catalyzed by SOS[54]. This markedly contrasts with Rasarfin, which we found inhibited both the SOS-dependent and -independent nucleotide exchange (e.g., induced by EDTA). This is likely due to the fact that DCAI only occupied a relatively small hydrophobic pocket (that involved interactions with Asp54, Ile55, Leu56, and Thr74 residues also shared by Rasarfin's benzofuran and halogenated aryl group), whereas Rasarfin covers almost the full length of the SOS-Ras interface on Ras. In addition, Rasarfin appears to lock switch I and switch II in an open-like conformation, which likely prevents Ras nucleotide binding and conformationally locks this Ras in a dysfunctional state[59]. Rasarfin's mode of action on Ras also

contrasts with the recently discovered molecule **3344** that interacts in a distinct hydrophobic pocket near the switch domains to block the small G protein's interactions with its effectors, or compound **22**, which bind directly SOS1 with high affinity to disrupts the K-Ras-SOS complex[57,60].

Our findings also suggest that Rasarfin inhibited ARF6 to block receptor internalization. Indeed, Rasarfin inhibited the recruitment of the clathrin adaptor AP-2 to β-arrestin2 and receptor internalization, both processes that require active ARF6[17]. However, the inhibitory mechanism seems to differ from Barbadin, which directly blocks β-arrestin/AP-2 interaction to inhibit receptor internalization[26], while Rasarfin likely indirectly prevents such complex formation via inhibition of AP-2 recruitment at the PM (e.g., limiting β-arrestin/AP-2 complexes assembly) and CCVs formation, which also requires ARF6[17]. The exact mechanism of Rasarfin interaction with this small G protein is unknown since structures of ARF6 with its GEFs are currently unavailable and a reliable model of Rasarfin binding on that small G protein cannot be deduced. Nonetheless, the switch I and II regions, which form the major sites for the interaction of ARF proteins with their cellular partners[61], suggest that Rasarfin may also bind in these regions. Moreover, despite the very conserved homology between ARF6- and ARF1-bound to GDP, the few sequence differences lead to important conformational differences between ARF1 and ARF6 switch regions[62], consistent with Rasarfin's selectivity on these small G proteins.

Rasarfin's inhibitory action on ARF6 also most likely differs from that of a recently described inhibitor, NAV-2729, since it was inferred to associate with ARF6 in its GEF-binding site based on a model of ARF1 in complex with one of its GEFs, ARNO (also referred to as Cytohesin[61]), and unlike Rasarfin, was inefficacious to block H-Ras[63]. Interestingly, similar to Ras-SOS interactions, the nucleotide exchange reactions on ARF promoted by ARNO also contribute, in part, to the dissociation of GDP through destabilizing bound $Mg^{2+}$ from the small G protein[59,64]. This is reminiscent of the way that Rasarfin may disrupt the divalent ion on Ras. Although we cannot exclude that Rasarfin targeted an ARF6 GEF, it is unlikely that it is ARNO because ARF1 activity would have also likely been affected[61]. Rasarfin is selective as it did not inhibit other small G proteins like Rho or Rac/Cdc42, which are closely related small G proteins and, in some cases, regulated by common GEFs[65,66]. Our findings are also consistent with observations from their respective structures with GEFs, suggesting that important residues in the hydrophobic groove between the switch domains of these small G proteins (e.g., Trp58 in RhoA, Trp56 in Rac, and Phe56 in Cdc42) induce a conformation that would interfere with Rasarfin binding if it was binding in this similar region as Ras[66]. Although we found that Rasarfin inhibited ARF6 activity and internalization, we cannot exclude that other small G proteins or effectors involved in receptor internalization/trafficking may also be affected by Rasarfin. More work is thus needed to better understand Rasarfin's inhibitory properties on ARF6, which may also aid in the design of more selective and/or efficacious endocytic and trafficking inhibitors.

Several studies have provided compelling evidence that signaling downstream of cell surface receptors controls internalization and intracellular trafficking of receptors, and that such regulations may vary between normal and cancer cells[7,8,67]. For instance, ERK1/2 has been reported to play a role in driving EGFR internalization in cancer cells through the regulation of CCP formation, while components of the MAPK signaling pathways were found to have functional interaction with the trafficking machinery[67,68]. Components of the MAPK signaling pathway, including ERK1/2, have also been shown to regulate β-arrestins' endocytic functions through phosphorylation during

the internalization and trafficking of some GPCRs[30,31,69]. Our findings, however, imply that signaling downstream of Ras may have played a minimal role in AT1R internalization in HEK293 cells. However, we cannot totally exclude that inhibition of receptor-mediated ERK1/2 by Rasarfin did not partially contribute to inhibiting the internalization and trafficking of other receptors or that in different cell contexts such regulation may be more obvious for AT1R. Indeed, because ERK1/2-mediated phosphorylation of β-arrestin increases receptor redistribution inside the cells, as well as delays in GPCR recycling to the PM through stabilization of receptor-β-arrestin complexes in endosomes, inhibiting MAPK with Rasarfin could have maintained a larger pool of receptor at the PM via increased receptor efflux, perhaps contributing partially to the observed lack of receptor removal from PM. However, such a possibility was not investigated here.

ARF6 has also been shown to play a role in AT1R-mediated ERK/12 activation in vascular smooth muscle cells where MAPK activity is high[70]. Here, in HEK293 cells, blocking ARF6 activity by expressing a DN of this small G protein, which partially inhibited receptor internalization, had little effect on MAPK activation. Notwithstanding such finding, we cannot totally exclude that ARF6 activity may not have been involved in MAPK signaling regulation through its activity on receptor internalization. Indeed, in β-arrestins-depleted cells, where preventing receptor internalization allows increased G protein-dependent MAPK signaling from the PM[23], ARF6-DN-mediated inhibition of AT1R internalization may maintain and/or increase ERK1/2 signaling at the PM with the result of failing to detect MAPK inhibition. On the other hand, Rasarfin, which also inhibited receptor internalization, efficiently blocked receptor-mediated ERK1/2 activation. Such latter effect, however, cannot be attributed to the inhibition of receptor internalization by Rasarfin since it also potently inhibited Ras. In that respect we did not observe persistence in AT1R-mediated MAPK signaling with Rasarfin when receptor internalization was inhibited as reported for other GPCRs[71], and as it would be expected if AT1R continued signaling via G proteins at the PM. For other GPCRs like the V2R, however, inhibition of internalization was sufficient to totally block ERK/12 activation[26]. Because Rasarfin prevented internalization and ERK1/2 through the respective inhibition of ARF6 and Ras, untangling the relative contribution of each small G protein in these responses, as well as the role of receptor trafficking in MAPK signaling, and vice-versa, will require further investigation. The development of Rasarfin derivatives with selective Ras vs. ARF6 inhibition properties should help address such questions.

Cancer cell proliferation is driven in part by altered intracellular signaling downstream of hyperactive growth factor receptors at the PM, including the engagement of the Ras-RAF-MEK-ERK and the PI3K/Akt pathways, as well as the crosstalk between signaling and endocytosis. Indeed, cancer cells are known to adapt their endocytic machinery to favorably promote their survival and progression, while signaling in these cells has also been observed to upregulate components of the endocytic and trafficking machinery[8]. While the prevalence of Ras in many cancers (e.g., through mutations, for instance) has been recognized for many years, ARF6 and/or its regulators has also been shown to be upregulated in some cancers[48,72]. For instance, ARF6, which not only controls trafficking in cells but also migration and invasion of cancer cells, is upregulated in some breast cancer tissues of high histological grades[73]. In that context, blocking both the internalization and trafficking of receptors, as well as their signaling through the combined inhibitory action on ARF6 and Ras, like Rasarfin does, may confer increased therapeutic advantages in some cancers. Our findings that Rasarfin

blocks Ras, MAPK, and ARF6 activities, and cell proliferation in breast cancer cells, for which no cancer therapeutics exist[74], support this idea.

## Methods

**Chemicals, reagents, and services.** Human AngiotensinII (AngII), Bradykinin (BK), Isoproterenol (ISO), epidermal growth factor (EGF), PD184352, Wortmannin, 3-(4,5- dimethylthiazol-2-yl)-2,5-diphenyltetrazolium bromide (MTT), Doxorubicin hydrochloride, poly-L-lysine hydrobromide, and poly-L-ornithine hydrochloride were purchased from Sigma Aldrich. UBO-QIC from Cedarlane. Dulbecco's modified Eagles medium (DMEM), fetal bovine serum (FBS), phosphate buffered saline (PBS), and gentamicin were purchased from Gibco, Life Technologies. Coelenterazine 400a (DeepBlue C) and Coelenterazine H were purchased from Nanolight Technology. Dimethyl sulfoxide (DMSO), phenylmethyl-sulfonyl fluoride (PMSF), leupeptin, aprotinin, pepstatin A, NaF, ampicillin, lysozyme, Triton-X, and EDTA were from BioShop. Glutathione Sepharose™ 4B was from GE healthcare. Linear polyethylenimine 25-kDa (PEI) was from Polysciences. The phospho-p44/42 MAPK (ERK1/2) (Thr202/Tyr204) (E10) (#9106, 1:1000), p44/42 MAPK (ERK1/2) (#9102, 1:2000), phospho-Akt (Thr308) (#9275, 1:1000), Akt (pan) (C67E7) (#4691, 1:1000), Ras (#3965, 1:1000) and RhoA (#2117, 1:1000) antibodies were purchased from Cell Signaling Technology. The anti-HA-Peroxidase (3F10) (#12013819001, 1:1000), anti-FLAG (#F7425, 1:1000) and anti-c-Myc (clone 9E10) (#M4439, 1:1000) antibodies were purchased from Sigma Aldrich. Anti-mouse (1:10,000) and anti-rabbit (1:10,000) IgG HRPs from BioRad. The β-actin (C4) (#sc-47778, 1:1000), H-Ras (#sc-520, 1:1000) and ARF6 (#sc-7971, 1:1000) antibodies were from Santa Cruz Biotechnology. SOS1 Protein (ExD Exchange Domain, aa564–1049, 6xHis tag) (#CS-GE02) and Mant-GTP exchange buffer (2X) (#EB01 from BK100 kit) were purchased from Cytoskeleton Inc.

**Compounds libraries.** The chemical library used for initial screening was from Chembridge DiverSet™ (60,000 compounds), Maybridge HitFinder™ (16,000 compounds), Maybridge (selected, 16,000 compounds), SPECS (selected, 16000 compounds), Microsource Discovery Spectrum (2000 compounds), Biomol (natural products, 500 compounds), Prestwick (commercialized products, 1120 compounds), SIGMA LopacTM (1280 compounds) and internal collection (synthesized at Université de Montréal-IRIC, ~4000 compounds). Compounds were all assigned an Université de Montréal number (UM) for internal use. Screen was performed at IRIC HTS facility[32]. KINOMEscan™ profiling was serviced by Eurofin/DiscoverX (San Diego, USA).

**Compounds acquisition.** Compounds **21** (Rasarfin) (CID: 1396167; Catalogue #001-728-363), **21.1** (CID: 2236635, Catalogue #001-615-578), **21.2** (CID: 2238454; Catalogue #001-009-312), **21.3** (CID: 2968266; Catalogue #001-728-365), **21.4** (CID: 1087127; Catalogue #001-728-361), **21.5** (CID: 1088362; Catalogue #001-629-834), **21.6** (CID: 1088375; Catalogue #001-629-837), **21.7** (CID: 2997077; Catalogue #002-020-863) and **21.8** (CID: 2944643; Catalogue #001-728-355) were purchased from MolPort and solubilized in 100% DMSO at a final stock concentration of 50 mM.

**Plasmids and constructs.** Plasmids encoding β-arrestin1-RlucII, AT1R-YFP[75], signal peptide-Flag tagged human AT1R (sp-Flag-AT1R)[76], β-arrestin2-YFP, HA-B2R, B2R-YFP[77], AT1R-RlucII, B2R-RlucII, β2AR-RlucII, HA-β2AR, rGFP-CAAX, rGFP-FYVE[32], β-arrestin2-RlucII, Gαi3-RlucII[78], Polycistronic Gαq sensor, Flag-Gβ1[32], GFP10-Gγ1[79], PKC, and Rho sensors[12], β-arrestin/AP-2 sensor[40,80] and GST-Rhotekin-RBD[81] were previously described. FLAG-K-Ras-WT, FLAG-K-Ras-G12V, FLAG-SOS1cat-CAAX were kindly provided by Dr. Matthew Smith (Université de Montréal, Qc). GST-Raf1-RBD, GST-GGA3-PBD, HA-ARF6, and HA-ARF6-T27N were described in refs. [70,73]. Myc-Rac1-WT, Myc-Rac1-Q61L, Myc-Rac1-T17N were kindly provided by Dr. Serge Lemay (McGill University, Qc).

To generate RlucII-tagged GGA3 (1–316) domain, the GGA3 (1–316) cDNA was amplified by PCR primers using GST-GGA3(1–316) DNA as a template. The PCR product was subcloned into the NheI/HindIII sites of RlucII containing vector (βarr2-RlucII[31] using Gibson assembly (New England Biolabs). Partial cDNAs of Raf and PAK were amplified by RT-PCR of HEK293 total RNA. For subcloning of the RBD of Raf, the RasBD was PCR amplified using the partial cDNA of Raf as a template and assembled into the NheI/HindIII sites of RlucII containing vector of βarr2-RlucII using Gibson assembly. The CRIB domain of PAK1 was PCR amplified using the partial cDNA of PAK1 as a template and subcloned into the KpnI/AgeI sites of RlucII containing vector (PKN-RBD-RlucII[12]), using Gibson assembly. For the Akt sensor, the PH domain of Akt was RT-PCR amplified from HEK293SL cell's total RNA as a template. The PCR product was reamplified with flanking sequences for the Gibson assembly. The final PCR product was assembled into NheI/HindIII sites of RlucII containing vector of βarr2-RlucII.

To generate pGEX-6P-1-H-Ras, the cDNA of H-Ras (full length) was obtained by RT-PCR of HEK293SL cell's total RNA. PCR product was then re-amplified with flanking sequences for the Gibson assembly. The final PCR product was subcloned into the pGEX-6P-1 vector, kindly provided by Dr. Matthew Smith (Université de Montréal, Qc), into the BamHI and NotI sites using Gibson assembly.

The S17N, Y32A, and Y40A substitutions in FLAG-K-Ras was generated by complementation PCR reaction, whereas the Y32A/Y40A substitution was generated by overlapping PCR amplification using the two PCR products for Y32A and Y40A as templates. All the final PCR products from the amplified complementing fragments were subcloned into XhoI/HindIII sites of FLAG-K-Ras vector using Gibson assembly. For generating the pGEX-6P-1-FLAG-K-Ras-WT and pGEX-6P-1-FLAG-K-Ras-Y32A, K-Ras-WT and K-Ras-Y32A DNA were PCR amplified and assembled into the BamHI/NotI sites of pGEX-6P-1 vector using Gibson assembly. All constructs were verified by DNA sequencing before use (McGill Genome Center). All the primers used are listed in Supplementary Table 1.

**Cell culture.** HEK293SL cells, characterized in ref. [82], are a subclone derived from regular HEK293 cells (Ad5 transformed) selected in our lab and have been used in all experiments. These cells have a cobblestone appearance and show better adherence as compared with regular HEK293 and HEK293T cells, making them more amenable to microscopy and BRET experiments. The MDA-MB-231 breast cancer cells were previously described in[83]. The A549 lung cancer cells were obtained from American Type Culture Collection (ATCC Cat# CRL-7909, RRID: CVCL_0023, Manassas, VA) and kindly provided by Dr. Jonathan Spicer (RI-MUHC, Montreal). CRISPR Gq/11 knockout cells were obtained from A. Inoue (Tohoku University, Sendai, Miyagi, Japan) and previously described in[12]. β-arrestin 1/2 KO cells were previously described in[23,28]. Cells were tested negative for mycoplasma contamination. (PCR Mycoplasma Detection kit, abm, BC, Canada) and cultured in DMEM supplemented with 10% FBS and 20 μg/ml of gentamicin. Cells were transiently transfected with conventional calcium phosphate methods or 25-kDa linear PEI (2:1 PEI/DNA ratio) methods.

**Live-cell imaging/confocal microscopy.** One day before transfection, cells were seeded in 35-mm glass-bottom dishes (MatTek Corporation) at a density of $1 \times 10^5$ cells per dish. For the recordings of receptor internalization, HEK293 SL cells were transfected with 2 μg of AT1R-YFP. For the recordings of β-arrestin-2 recruitment to the receptor, HEK293SL cells were transfected with 50 ng of β-arrestin2-YFP and 250 ng of Flag-AT1R or HA-B2R. Forty-eight hours post-transfection, cells were serum starved, preincubated with DMSO (0.1% final concentration) or Rasarfin (50 μM) for 30 min at 37 °C. AT1R-expressing cells were stimulated with angiotensinII (AngII; 100 nM) and B2R-expressing cells were stimulated with bradykinin (BK; 1 μM) for 30 min (receptor-YFP) or 15 min (β-arrestin2-YFP). Cells were imaged with Zeiss LSM-510 and/or LSM-710 laser scanning confocal microscope. To detect YFP, UV laser was used with 405 nm excitation and BP 505–550 nm emission filter. Images (2048 × 2048) were collected using a 63x oil immersion lens.

**BRET measurements.** HEK293SL cells were seeded at a density of $7.5 \times 10^5$ cells per 100-mm dish and 24 h later, transiently transfected as such. For receptor internalization experiments, cells were transfected with 0.12 μg AT1R-RlucII, B2R-RlucII, or β2AR-RlucII and 0.48 μg of either rGFP-CAAX (to assess removal from the PM) or rGFP-FYVE (to assess accumulation in the endosomes). For β-arrestin recruitment assays, cells were transfected with 0.48 μg of receptor-YFP along with 0.12 μg of β-arrestin-RlucII. For β-arrestin/AP-2 binding experiments, cells were transfected with 1 μg Flag-AT1R, 1 μg β2-Adaptin-YFP, and 0.12 μg β-arrestin2-RlucII. For G protein activation, cells were transfected with 3 μg of sp-Flag-AT1R along with either 4.5 μg of the Gαq-polycistronic BRET sensor or 0.24 μg of the Gαi3-RlucII and 0.6 μg of GFP10-Gγ2 and Gβ1 sensors or 0.12 μg PKN-RBD-RlucII and 0.48 μg of rGFP-CAAX. For PKC activation, cells were transfected with 3 μg of sp-Flag-AT1R and 0.18 μg of the PKC sensor. For GTPase activation experiments, cells were transfected with 1 μg Flag-AT1R, 0.48 μg of rGFP-CAAX, and either 0.12 μg of GGA3-PBD-RlucII, PAK-CRIB-RlucII or Raf-RBD-RlucII. For biosensor validation experiments, cells were additionally transfected with 500 ng FLAG-K-Ras-WT, Flag-K-Ras-G12V, Flag-K-Ras-SOS-CAAX, HA-ARF6-WT, HA-ARF6-T27N, Myc-Rac1-WT, Myc-Rac1-Q61L or Myc-Rac1-T17N. For PI3K/Akt activation experiments, cells were transfected with 0.48 μg of rGFP-CAAX and 0.12 μg of Akt (PH)-RlucII. After 18 h of transfection, the media was replaced and cells were divided for subsequent experiments. Cells were detached and seeded onto poly-L-ornithine-coated 96-well flat white bottom plates (BrandTech Scientific) at a density of $2.5 \times 10^4$ cells per well in media. The next day, cells were washed once with Tyrode's buffer (140 mM NaCl, 2.7 mM KCl, 1 mM CaCl2, 12 mM NaHCO3, 5.6 mM D-glucose, 0.5 mM MgCl2, 0.37 mM NaH2PO4, 25 mM HEPES, pH 7.4) and left in Tyrode's buffer. For kinetics of β-arrestin recruitment or β-arrestin binding to receptor and AP-2 experiments, cells were serum starved for 30 min, pretreated with **21** (50 μM) or Barbadin (100 μM) for 30 min, stimulated with 100 nM AngII or 1 μM BK and BRET signals were monitored at indicated times using a Victor X Light plate reader (Perkin-Elmer). Coelenterazine H (final concentration of 5 μM) was added 3–5 min prior to BRET measurements. Filter set was 460/80 nm and 535/30 nm for detecting the RlucII *Renilla* luciferase (donor) and YFP (acceptor) light emissions, respectively. The BRET ratio was determined by calculating the ratio of the light emitted by YFP

over the light emitted by the RlucII. For G protein activation, cells were serum starved, pretreated with compounds for 30 min, stimulated with 100 nM AngII for 2 min ($G\alpha_q$, $G\alpha_{i3}$, and Rho sensors) or 5 min (PKC sensor). For the kinetics of GTPase and PI3K/Akt activation, cells were serum starved, pretreated with **21** (50 µM) or Wortmannin (200 nM) for 30 min, stimulated with ligand (100 nM AngII or 100 ng/ml EGF) and BRET signals were monitored at indicated times. For concentration-response curves, cells were serum-starved, pretreated with various concentrations of compounds for 30 min and stimulated with ligand (100 nM AngII, 1 µM BK, 10 µM Isoproterenol (Iso) or 100 ng/ml EGF) in Tyrode's buffer for 30 min for receptor internalization, 10 min for ARF6, 2 min for Rac and Rho activation and 5 min for Ras activation and PI3K/Akt activation. BRET signals were monitored using a Synergy2 (BioTek) microplate reader and coelenterazine 400a (final concentrations of 5 µM) added 3–5 min prior to BRET measurements. Filter set was 410/80 nm and 515/30 nm for detecting the RlucII *Renilla* luciferase (donor) and rGFP/GFP10 (acceptor) light emissions, respectively. The BRET ratio was determined by calculating the ratio of the light emitted by rGFP/GFP10 over the light emitted by the RlucII.

**Western Blot analysis.** HEK293SL cells ($10^5$ cells per well) were seeded in a poly-L-lysine-coated six-well plate and transiently transfected with 3 µg Flag-AT1R, HA-B2R or HA-β2AR and/or 500 ng Flag-MEK1-WT, Flag-MEK1-DD, Flag-BRAF-WT, Flag-BRAF-V600E, Flag-K-Ras-WT, Flag-K-Ras-G12V, HA-ARF6-WT, or HA-ARF6-T27N. $G_{\alpha q/11}$ KO and βarr1/2 KO cells were seeded onto 100 mm dishes at a density of $10^6$ cells per dish. Next day, cells were transfected with 3 µg AT1R alone or along with 500 ng of β-arrestin2 using PEI. After 18 h of transfection, cells were detached and re-seeded onto poly-L-lysine coated 12-well plates (~1.5–2 × $10^5$cells/well) for subsequent experiments. Forty-eight hours post transfection, in a 37 °C water bath, cells were serum-starved for 30 min, pretreated with DMSO (0.1%) or compound **21**/Rasarfin (50 µM or at indicated concentrations) for 30 min, then stimulated or not with the indicated ligand [AngII (1 µM), BK (1 µM), Iso (10 µM) or EGF (100 ng/ml)] at indicated times. Cells were put on ice, washed with PBS and solubilized in 2x laemmli buffer (250 mM Tris–HCl pH 6.8, 2% SDS (w/v), 10% glycerol (v/v), 0.01% bromophenol blue (w/v) and 5% β-mercaptoethanol (v/v)) by heating at 65 °C for 15 min. Lysates were resolved on 10 or 14% SDS-PAGE, transferred to nitrocellulose membranes and immuno-blotted for p-ERK1/2, ERK1/2, p-Akt, Akt, HA, Myc, β-actin or FLAG. ImageLab 5.2 software was used to quantify the digital blots as fold of the phosphorylated protein over total protein, which were then normalized as indicated in the figure legends. Uncropped immunoblots are provided in the Source Data file.

**Purification of recombinant protein.** GST, GST-tagged Golgi Associated, Gamma Adaptin Ear Containing, ARF Binding Protein 3 Binding Domain (GST-GGA3-PBD), Raf1-Ras Binding Domain (GST-Raf1-RBD) and Rhotekin-Rho Binding Domain (GST-Rhotekin-RBD), as well as Ras proteins (pGEX-6P-1-H-Ras, pGEX-6P-1-K-Ras-WT and pGEX-6P-1-K-Ras-Y32A) were expressed in E. coli BL21 cells and grown overnight in LB medium (25 g/l) with ampicillin (100 µg/ml). Cells from overnight culture (3 ml) were then transferred to 250 ml medium in a 1 L volu-metric flask and induced with isopropyl β-D-1-thiogalactopyranoside (IPTG) under the respective conditions: 1 mM for 1 h at 37 °C (GST and GST-Raf1-BD), 0.6 mM for 3–4 h at 30 °C (GST-Rhotekin-RBD and GST-GGA3-PBD) or 0.12 mM for 16 h at 15 °C (pGEX-6P-1-H-Ras, pGEX-6P-1-K-Ras-WT, and pGEX-6P-1-K-Ras-Y32A). The 250 ml culture was transferred into 50 ml tubes and centrifuged for 15 min at 4000 rpm. On ice, the pellet was resuspended with 3 ml/tube Binding Buffer (2 mg/ml lysozyme, 10% glycerol (v/v), 50 mM Tris-HCl pH 7.8, 100 mM NaCl, 1% Nonidet P-40 (v/v), 1 mM dithiothreitol, 1 mM EDTA) supplemented with protease inhibitors (1 mM phenylmethyl-sulfonyl fluoride (PMSF), 10 µg/ml leupeptin, 5 µg/ml aprotinin, and phosphatase inhibitors (20 mM NaF), mixing all tubes in one. The tube was placed on ice for 15 min, 1% Triton-X was then added and frozen in liquid nitrogen. It was thawed once at 37 °C then placed back on the ice and sonicated for 5 s on ice. Aliquots were centrifuged at 13,000 × g for 10 min at 4 °C, the supernatant was transferred to a new tube on ice and pellet discarded. Glutathione Sepharose 4B beads were centrifuged at 10,000 × g for 1 min and supernatant discarded. Beads were washed twice with Binding Buffer. A 50% slurry of beads were added to the supernatant of GST fusion protein and rotated for 1 h at 4 °C. GST proteins coupled to glutathione resin were then centrifuged at 10,000 × g for 2 min and supernatant discarded. Beads were washed twice with Binding Buffer and resuspended on ice with a 50% slurry volume of Binding Buffer and stored at 4 °C until use. Uncropped coomassie gels are provided in the Source Data file.

**Glutathione S-transferase (GST) pull-down assays.** Activation of ARF1, ARF6, Ras, and Rho were assessed by using GST pull-down assays. HEK293SL cells were transiently transfected with 3 µg AT1R-flag only (Ras and Rho) or along with 500 ng HA-ARF6 (ARF6) or HA-ARF1 (ARF1). Forty-eight hours later, the HEK293 or MDA-MB-231 cells were serum-starved for 4 h with DMEM con-taining 20 mM HEPES then pretreated with DMSO or Rasarfin (50 µM) for 30 min. Cells were then stimulated with 1 µM AngII or 100 ng/ml EGF for 0 and 2 min. Cells were then washed once with ice-cold PBS and lysed for 30 min at 4 °C in 300 µl of lysis buffer (pH 7.5, 50 mM Tris-HCl, 140 mM NaCl, 5 mM $MgCl_2$, 10% glycerol (v/v), 1% Nonidet P-40 (v/v), 1 mM dithiothreitol) supplemented with protease inhibitors (1 mM phenylmethyl-sulfonyl fluoride (PMSF), 10 µg/ml leu-peptin, 5 µg/ml aprotinin, 1 µg/ml pepstatin A) and phosphatase inhibitors (20 mM NaF, 0.025 mM pervanadate). The samples were cleared by centrifugation and 30 µl (cell lysates) was kept for assessing total protein contents. The remaining was transferred to fresh tubes with 20 µg of either GST, GST-GGA3-PBD, GST-Raf1-RBD or GST-Rhotekin-RBD coupled to glutathione resin and rotated for 1–2 h at 4 °C. Beads were washed twice with lysis buffer and proteins were eluted in 25 µl 2x laemmli buffer by heating at 65 °C for 15 min. Proteins were resolved on 14% SDS-PAGE, transferred to nitrocellulose membranes, and immunoblotted for HA, Ras, RhoA, H-Ras, or ARF6. ImageLab 5.2 software was used to quantify the digital blots as fold of the amount of pulled down protein over total protein. Uncropped coomassie gels and immunoblots are provided in the Source Data file.

**GTP exchange factor (GEF) assay.** Ras activity was assessed in a 384-well black bottom plate using the mant-GTP exchange factor assay that measures the uptake of the fluorescent nucleotide analog N-methylanthraniloyl-GTP (mant-GTP) into GTPases. For Ras activation, 2X exchange buffer, 1.66 µg of purified pGEX-6P-1-H-Ras or pGEX-6P-1-K-Ras-WT or pGEX-6P-1-K-Ras-Y32A were added per well in presence of DMSO, different concentrations of Rasarfin (as indicated) or 50 µM of compounds **21.4**, **21.7**, or **21.8**. Using the Infinite 200 Pro plate reader (Tecan) with filters set at 360 nm and 440 nm for detecting the excitation and emissions, respectively, and temperature set at 20 °C, 5 readings were recorded every 5 s before the addition of either $H_2O$, 40 mM EDTA or 0.66 µg purified SOS1. The fluores-cence of mant-GTP uptake was measured every 30 sec for 30 min and quantified as the delta relative fluorescence units (RFU), which was calculated as the RFU post-addition minus the 5 averaged RFU pre-addition, per condition.

**Computational site finding and docking of Rasarfin and compound 21.4.** SOS1 (chain S) and water molecules were deleted from the SOS-Ras X-ray structure (PDB 1BKD). The Ras-structure (chain R) was protonated and charged accordingly using Structure Preparation in Molecular Operating Environment (MOE) 2019.01.04 (Chemical Computing Group ULC, Montreal, QC, Canada, 2019). The Site Finder Tool in MOE suggests a binding site of 47 side chain contact atoms on the protein. The selected binding site has a PLB (propensity of ligand binding)[84] of 1.77 and displays 25 hydrophobic contact atoms on the protein. Dummy atoms were used as a binding site reference for the flexible protein docking procedure. Triangle Placement was carried out using Triangle Matcher, which allows 300 s for ligand placement and a maximum of 1000 poses returned for the compound. London dG was used for the Scoring as it estimates the free Energy of binding for each pose. Post-placement refinement was carried out employing Induced Fit of the Receptor, and the Force Field-based Scoring Function GBVI/WSA dG for final Scoring, retrieving 10 docking poses for each compound. Docking poses were evaluated according to their docking scores, protein-ligand interaction fingerprints (PLIFS), and visual inspection. We selected the highest scoring pose which exhibited the most PLIFs resembling the protein–protein interactions of SOS1 and Ras.

**Molecular dynamics simulations.** To simulate Ras protein in complex with **21**/ Rasarfin, compound **21.4**, as well as in the apo state, we used structures generated in the docking step. For Rasarfin association simulation, five Rasarfin molecules were placed at random in the vicinity of the protein. The structures were solvated using TIP3P waters in CHARMM-GUI (a web-based graphical user interface for CHARMM). Ionic strength of the systems was kept at 0.15 M using NaCl ions. Protein parameters were obtained from the CHARMM36m (an improved force field for folded and intrinsically disordered proteins) and Charmm36 (Updated version of the CHARMM all-atom additive force field for lipids: validation on six lipid types), respectively. Ligand parameters were automatically assigned using Automation of the CHARMM General Force Field (CGenFF) I (bond perception and atom typing) and Automation of the CHARMM General Force Field (CGenFF) II (assignment of bonded parameters and partial atomic charges) from the CGenFF forcefield (a force field for drug-like molecules compatible with the CHARMM all-atom additive biological force field), which is an extension of the CHARMM General Force Field to sulfonyl-containing compounds and its utility in biomolecular simulations.

Generated systems were simulated using the ACEMD (accelerating biomolecular dynamics in the microsecond time scale). The simulation protocol included an equilibration step of 50 ns in condition of constant pressure (1.01325 bar - NPT). During this step the backbone of the receptor as well as the ligand were constrained, and the timestep was set at 2 fs. The pressure was kept constant using the Berendsen barostat [Molecular dynamics with coupling to an external bath]. This step was followed by 3 production runs of 4 µs for the Rasarfin-bound, **21.4**-bound, as well as the apo state of Ras protein. To simulate Rasarfin association we carried out 3 production runs of 1 µs. The production runs were done using a timestep of 4 fs in conditions of constant volume (NVT). During both NPT and NVT runs, temperature was maintained constant using the Langevin thermostat (MD simulation for polymers in the presence of a heat bath). Van der Waals and short-range electrostatic interactions were set with a cut-off of 9 Å and a switching potential applied at 7.5 Å, long-range electrostatic interactions were approximated using the Particle Mesh Ewald method (an N·log(N) method for

Ewald sums in large systems). Analysis of systems was carried out using VMD (visual molecular dynamics). All volumetric occupancy maps were plotted at isovalue of 0.15.

**MD pharmacophore analysis**. The starting coordinates and trajectories of the protein and the respective compound bound were imported into LigandScout 4.4.1 (Inte:Ligand GmbH)[85] using a stride of 50. The "MD Pharmacophores" tool was employed to create dynamic pharmacophores to show protein-compound interaction patterns and to furthermore calculate their frequency. Calculated interactions were based on the following distances and angles: Hydrophobic interactions: 1–5 Å. H-bond donor/acceptor: 2.5–3.8 Å; Angle tolerance of 180° for $sp^3$ hybridized atoms is an ideal hydrogen bond, which is broken when the angle difference exceeds 34° in either direction around the central position (angle tolerance of 50° is allowed for $sp^2$ hybridized atoms); Angle tolerance of 60° for pi-cation interactions; Angle tolerance of 20° for orthogonal pi-pi interactions and 20° for parallel pi-pi interactions; Aromatic interactions: 0.0–2.0 Å orthogonal/parallel center deviation (minimum and maximum distance of two orthogonal or parallel plane feature center points). The final output displays the number of unique pharmacophores, appearance frequency, and Feature Timeline of computed interactions between the compound and the protein.

**Cell proliferation assay**. Cell proliferation was measured by a label-free, non-invasive cellular confluence assay using IncuCyte Live-Cell Imaging Systems (Essen Bioscience, Ann Arbor, MI, USA). MDA-MB-231 breast cancer and A549 lung carcinomas cells (1500 cells in 150 µl/well) were seeded overnight on a clear 96-well plate. Twenty-four hours later, the cells were treated with 0.1% DMSO, or different concentrations of Rasarfin (as indicated), or 10 µM of **21.4** or **21.8** for 96 h total. The plate was placed in an XL-3 incubation chamber maintained at 37 °C and the cells were photographed using a ×4 objective, every 3 h for 4 days. Cell confluence was calculated using IncuCyte S3 software (2019A) and cell proliferation was expressed as an increase in the percentage of confluence. Experiments were done in triplicate and repeated three times.

**Cell viability assay**. MDA-MB-231 breast cancer cells and A549 lung carcinomas (1500 cells in 150 µl/well) were seeded overnight on a clear 96-well plate. Twenty-four hours later, the cells were treated with 0.1% DMSO, or different concentrations of Rasarfin (as indicated), or 10 µM of **21.4**, **21.8** (MDA-MB-231 breast cancer cells) or Doxorubicin for the indicated times at 37 °C. At each time point, 10 µl of 5 mg/ml MTT was added to each well and the plates were incubated for an additional 4 h at 37 °C. The absorbance at 590 nm was measured by plate reader. Experiments were done in triplicate and repeated three times.

**Data analysis**. Statistical analyses were performed using GraphPad Prism 6 software (GraphPad Software Inc.; La Jolla, CA) using either two-tailed unpaired Student's t-tests, one-way or two-way ANOVAs, corrected with Bonferroni or Dunnett's comparisons tests, when appropriate and as indicated in the figure legends. Curves presented throughout this study were generated using GraphPad Prism software and represent the best fits, from which $IC_{50}$s were calculated. $P$ values < 0.05 were considered significant. Figures were generated using Adobe Illustrator (2020) and chemical structures using ChemDraw (1.9.3).

**Reporting summary**. Further information on research design is available in the Nature Research Reporting Summary linked to this article.

## Data availability
The main data supporting the findings of this study are presented within the article and its Supplementary Information files, and are available from the corresponding author upon reasonable request. The source data underlying Figs. 1–4, 8, Supplementary Figs. 1–6 and Supplementary Figs. 11–12 are provided as a Source Data file. Specific data P values are also included within the Source Data file. All compounds for the library screen have been codified with a University of Montréal number (UM) for internal use. UM corresponding commercial compounds' catalogue numbers, which include structures, are available upon request. Information about IRIC internal compounds' collection can be obtained with a disclosure agreement from A.M. (anne.marinier@umontreal.ca). Source data are provided with this paper.

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

## Acknowledgements

We thank Dr. Min Fu and the Molecular Imaging Platform of the Research Institute of the McGill University Health Centre (RI-MUHC) for assistance with microscopy experiments. We thank Christian Le Gouill at Institute for Research in Immunology and Cancer (IRIC) for discussions about the design of the BRET biosensors. We thank Yubo Cao for the illustration of biosensors. T.M.S. is supported by the National Science Centre of Poland, project number 2017/27/N/NZ2/0257. J.S. is supported by the Instituto de Salud Carlos III FEDER (PI15/00460 and PI18/00094) and the ERA-NET NEURON & Ministry of Economy, Industry, and Competitiveness (AC18/00030). This work was supported by grants from the Canadian Institutes of Health Research (CIHR) to S.A.L. (MOP74603) and to M.B. (FDN148432), and a team grant from the Réseau québécois de recherche sur le médicament (RQRM) to S.A.L. and M.B. M.B. holds a Canada Research Chair in Signal Transduction and Molecular Pharmacology. M.L.M. is supported by a doctoral studentship from CONACyT.

## Author contributions

J.G., M.B., and S.A.L. designed the study. J.G. designed, performed, and analyzed BRET, MAPK, pulldown, microscopy, and in vitro experiments. D.A.S., A.B., T.M.S., J.S., and A.M. designed, performed, and analyzed the computational data. Y.N. designed and generated the BRET sensor constructs and characterized them with J.G. E.K. designed and performed microscopy experiments. J.G. and M.L.M. designed and performed cell proliferation and viability assays and analyzed data with N.L.V. S.C. and A.C. designed and interpreted small G proteins biochemical assays. J.D. S.A., and O.R. performed and validated the HTS. J.G. and Y.N. analyzed HTS data. All authors reviewed the data. J.G. and S.A.L. wrote the initial manuscript, and Y.N., D.A.S., A.M., A.C., N.L.V., and M.B. contributed to the final version. S.A.L. supervised the project.

## Competing interests

Some of the BRET-based biosensors used in the present study are licenced to Domain Therapeutics for commercial use. The biosensors are freely available under material transfer agreement for academic research and can be requested from S.A.L. (stephane. laporte@mcgill.ca) or M.B. (michel.bouvier@umontreal.ca). M.B. is the president of the Domain Therapeutics Scientific Advisory Board. All other authors declare no competing interests.
