## [Peer Review File · Nature Communications]

Reviewers' Comments:

Reviewer #1:

Remarks to the Author:

Giubilaro and colleagues report on the Discovery of a dual Ras and ARF6 inhibitor from a GPCR endocytosis screen. They name the novel compound rasarfin and find that it blocks agonist-mediated internalization of the angiotensin II AT1 receptor and of other GPCRs as well as signaling of GPCRs and RTKs. They correlate its effect with the inhibition of cancer cell proliferation and therefore claim that the novel compound may be useful for inhibition of oncogenic signaling. Yet the authors only show effects of rasarfin in a single cancer cell line. The paper is well written with very few flaws only and does contain some interesting information and an inhibitor with an interesting mechanism of action. Despite these positive attributes, this reviewer notes some some imbalanced statements on GPCR signaling mechanisms citing quite a number of 'old' papers while omitting much more recent and relevant literature. In addition, not all claims are supported by data, so I have made specific suggestions how to overcome these shortcomings and the semantic imprecisions. Finally, it is not clear to me if there is a mechanistic link between inhibition of trafficking and inhibition of signaling or whether these phenomena are accidentally correlated. Specific experiments are missing to clarify this important issue. Please see my comments below for more information...

Major Points:

- Introduction, lines 45,46: "Dysregulation of receptor's internalization and signaling underly the aberrant cell responsiveness found in some cancers." Please provide some very specific references showing that dysregulation of receptor internalization is indeed linked to cancers. Dysregulation of receptor signaling has strong links to cancer but the evidence for aberrant internalization must be included here.

- Introduction, lines 54-58 "For instance, trafficking of β -arrestins into endosomes with GPCRs like the angiotensin II type 1, bradykinin B2 and vasopressin V2 receptors (AT1R, B2R, and V2R, respectively), engages receptors in mitogen-activated protein kinases (MAPKs) signaling 12-16,...please provide a more balanced view and include references to much more recent literature clearly showing that signaling of AT1R and V2 does not only require arrestins to engage MAPK signaling from endosomes but occurs in the absence of arrestins and any internalization from the PM as well.

This sentence continues... while β -arrestins mediated ERK1/2 for the B2R, V2R and β -adrenergic receptors (β 1AR and β 2AR) have been shown to also occur at the PM 17-19." Please be more precise and clearly discriminate whether you refer to G protein independent arrestin-mediated ERK or G protein-dependent arrestin-modulated ERK. The term beta arrestin-mediated ERK is semantically imprecise and potentially misleading, particularly as recent literature has shown that arrestins are dispensable for initiation of ERK signaling for both beta2AR and V2R. Reference to this much more recent literature must be made here and the term arrestin-mediated ERK must be avoided unless arrestin's role as transducer or scaffold is clearly stated.

- Introduction, lines 64-67: "Therefore, the development of new pharmacological tools that block receptor internalization at distinctive steps would not only provide a unique opportunity for addressing the role of β -arrestins-mediated MAPK signaling in different cell compartments, but to also control specific mitogenic responses in cells." The authors make important claims here in that internalization inhibitors would allow to adress the role of arrestins for MAPK signaling. They identify an inhibitor of internalization and this molecule also inhibits MAPK signaling. But, most importantly, the authors make no effort in this manuscript to show that these processes are intertwined. Thus they MUST examine whether their novel inhibitor rasarfin inhibits MAPK signaling in cells lacking barr1 and barr2, where AT1R does not internalize. If rasarfin looses its effect on MAPK signaling, the links they attempt to make would hold. At present, the link is unsupported. Barbadin, previously published and

included in the report, must be used as control under the exact same conditions in barr1/2 depleted cells. both ligands should lose their effect when arrestins are genetically absent. Such data are very simple to generate and of high mechanistic importance to support the above links between trafficking and signaling.

- lines 71-73: AT1R engages MAPK signaling in endosomes... add newer literature to mention that it also signals from the PM in the absence of any internalization. This has clearly been demonstrated but has been omitted so far.

- line 76: "receptor/ β -arrestin complex signaling at the PM" please define what this is. which type of signaling do you refer to? G protein-independent signaling or arrestin-contribution to G protein dependent signaling? As there is plenty of misleading literature in this field, please be semantically precise to avoid further confusion.

Results: Figure 1a and Supplementary Table 1. The table is not really helpful as it provides no information to the interested reader. Instead I would request to show the structures of the compounds listed in Fig1b. Only compound 21 is shown, the other active and inactive compounds are also interesting from a structural perspective. Are there analogs with similar profiles? Families of active compounds? All this information is hidden.

Same is true for Fig1b and Suppl Table 2, it has no value without structures. Readers must see how selected compounds look like.

the selection of compound 21 in Fig1 is too cryptic. Low cell toxicity must be shown for this compound. If cell rounding was analyzed, how does it look like? Why was 21 chosen and not another one that inhibits the BRET signal better? Please consider indicating which of the compounds are toxic at 50 micromolar already in this table in Fig1b. The selection of cpd 21 requires more transparency.

Fig1e: the concentration of compound 21 must be given in the legend. This is important to allow reproduction of data by others.

data related to Fig1: please explain why you omitted barr1 in Fig1f. Does cpd 21 inhibit barr1 recruitment to B2R? Was this tested?

Fig 2a: what is the concentration of 21? this must be stated. Barbadin is used as reference here at very high molar concentration (100 micromolar). Does Barbadin at this very high concentration induce any toxic effects to cells visible for example in cell rounding? As concentrations are not given for 21 and no statement is made whether the applied concentrations of both compounds are maximally effective, quantitative comparison of arrestin-AP2 complex formation is not permitted. Lacking information must be provided to make sure conclusions are supported by data.

Are the data shown in Supp Table3 n=1? The number of technical and biological replicates must be given. If this is a single data point only, it must be stated clearly in the legend as such data are associated with quite some uncertainty.

Minor Points:

Abstract: "Internalization and intracellular trafficking of hormone receptors, like receptor tyrosine kinases (RTKs) and G protein-coupled receptors (GPCRs), play pivotal roles in cell responsiveness homeostasis." Please check this sentence, is it complete?

Introduction, line 44: "maintain cell responsiveness homeostasis". Is this sentence complete?

line 176: pathways instead of patways

line 204: also instead of alos

Reviewer #2:

Remarks to the Author:

The manuscript "Discovery of dual Ras and ARF6 inhibitor from a GPCR endocytosis screen" by Giubilaro et al. presents results of a HTS, biochemical and in silico work to identify a dual RAS/ARF6 inhibitor.

Identification of Rasarfin was based on HTS and validated by several biochemical assays and supported by in silico methods (docking, MD simulations).

I have some very serious concerns over the validity of in silico work. I will analyze here the most critical aspects, without going into the details.

1. Modeling of hRAS: Docking. There is no data (except one image in Figure 5 and one supplementary image) showing the individual docking poses. Especially one would need information if the poses of all compounds do like each other and if individual poses are similar (within the same compound). It is well known that scoring functions do not yield reliable predictions concerning the docking poses, so one must use quite a lot subjective analysis.

2. MD simulations: This is a critical issue. All the used simulations are absolutely too short to show any reliable information if hit compounds are binding hRAS or not. Authors should look at two papers how MD simulations should be done and especially how those should be analyzed

-(Grossfield A, Patrone P, Roe D, Schultz A, Siderius D, Zuckerman D (2018) Best Practices for Quantification of Uncertainty and Sampling Quality in Molecular Simulations [Article v1.0]. Living Journal of Computational Molecular Science 1: . <https://doi.org/10.33011/livecoms.1.1.5067>) AND

-(Braun E, Gilmer J, Mayes H, Mobley D, Prasad S, Zuckerman D, Monroe J (2018) Best Practices for Foundations in Molecular Simulations [Article v1.0]. Living Journal of Computational Molecular Science 1: . <https://doi.org/10.33011/livecoms.1.1.5957> FOR YOUR INFORMATION, THERE IS NO NEED TO ADD THESE AS A CITATION).

In the current manuscript there is no analysis of the quality of MD (RMSD is not enough here, look the references above!). Also, if you look at the typical timeframe how long simulations are needed for RAS it is clear that at least a microsecond scale is needed. In a kRAS study -(Pantsar T, Rissanen S, Dauch D, Laitinen T, Vattulainen I, Poso A (2018) Assessment of mutation probabilities of KRAS G12 missense mutants and their long-timescale dynamics by atomistic molecular simulations and Markov state modeling. PLoS Computational Biology 14: . <https://doi.org/10.1371/journal.pcbi.1006458>)- simulations were at the level of 5 microsecond/protein. Also, another good example for kRAS MD studies -(Vatansever S, Erman B, Gümüş ZH (2019) Oncogenic G12D mutation alters local conformations and dynamics of K-Ras. Scientific Reports 9:11730 . <https://doi.org/10.1038/s41598-019-48029-z>)- used a microsecond scale simulations.

The reason why I also believe these long simulation times are mandatory is coming from the nature of proposed binding mode. The putative binding pocket is in between Switches 1 and 2. Both of the previously mentioned studies found out that exactly these domains are quite flexible . Since the used X-ray structure is based on the SOS1/hRAS complex it has a lot of structural strain due to the existing SOS1 interaction. Once this is removed it is clear that very long simulations are needed before a realistic binding mode can be achieved. Used ns-scale simulations are not more than 1-5% of what is needed. I am aware that this requires a huge computational power, but if this is not done then one cannot claim that MD is indicating anything about the binding.

3. Pharmacophore analysis should be validated either by searching new compounds OR using some external compounds in the dataset and checking if predictions are working. However, in this case a more robust approach would require simulations of both known active and inactive analogs with hRAS and checking if MD is able to discriminate compounds.

4. Why not ARF6 modeling at all? It is quite probably that binding mode is similar between hRAS and ARF6.

A minor comment: please check the simulation videos (supplementary material), at least one simulation includes extra material (menu down right corner) and a suspicious jump in the middle of trajectory.

REVIEWER COMMENTS AND ANSWERS

We thank both reviewers for their careful assessment of our work and their constructive comments. Below are our answers to the reviewers' questions. Modifications to the text can also be found in the red-track, line-numbered version of the manuscript. We believe we addressed all issues, and that our manuscript has significantly improved.

Reviewer #1 (Remarks to the Author)

“The paper is well written with very few flaws only and does contain some interesting information and an inhibitor with an interesting mechanism of action.”

We thank this reviewer for his/her encouraging comments, especially about it containing interesting information, and for helpful suggestions to improve the present work.

“They correlate its effect with the inhibition of cancer cell proliferation and therefore claim that the novel compound may be useful for inhibition of oncogenic signaling. Yet the authors only show effects of rasarfin in a single cancer cell line”

We agree with this reviewer that Rasarfin's effect was only shown in one cancer cell line and therefore added new data using a second cell line: the lung carcinomas A549 cells which are commonly used for both basic research and drug discovery and known to have increased Ras and EGFR activity¹. We now show with these cells that Rasarfin efficiently blocks proliferation of lung carcinomas cells, similarly to what was observed in MDA-MB-231 breast cancer cells (Supplementary Fig.12 and described the results lines 425-430).

“This reviewer notes some some imbalanced statements on GPCR signaling mechanisms citing quite a number of 'old' papers while omitting much more recent and relevant literature”

This reviewer is correct that some citations appeared older than others, creating what he/she describes as unbalanced statements. It is, however, our viewpoint to cite as much as possible the first demonstration of the phenomena/concepts we refer to. For more universal and well accepted concepts, we generally cite reviews, also considering ones from different periods. However, at the reviewer's suggestion, we added more contemporary references, about different pathways for GPCR-mediated MAPK activation (see also comments below).

“Finally, it is not clear to me if there is a mechanistic link between inhibition of trafficking and inhibition of signaling or whether these phenomena are accidentally correlated”

This reviewer is correct about such lack of definitive conclusion about a causal link between receptor trafficking and signaling, since our new compound (**21** a.k.a. Rasarfin) affects both processes by targeting ARF6 to block internalization and Ras to inhibit MAPK signaling. This is now extensively addressed in the discussion and we propose new research avenues to further explore this (lines 519-544).

Major Points:

Introduction, lines 45,46: "Dysregulation of receptor's internalization and signaling underly the aberrant cell responsiveness found in some cancers." Please provide some very specific references showing that dysregulation of receptor internalization is indeed linked to cancers. Dysregulation of receptor signaling has strong links to cancer but the evidence for aberrant internalization must be included here.

We have now revised the text and added supported reviews for RTKs for which dysregulation of receptor trafficking and signaling have been well documented. We also better contextualize GPCR dysregulation of signaling in cancer, and report reviews for which perturbation of the endocytic trafficking machinery has been discussed to be linked to oncogenic responses (lines 49-55)²⁻⁴.

Introduction, lines 54-58 "For instance, trafficking of β -arrestins into endosomes with GPCRs like the angiotensin II type 1, bradykinin B2 and vasopressin V2 receptors (AT1R, B2R, and V2R, respectively), engages receptors in mitogen-activated protein kinases (MAPKs) signaling 12-16,...please provide a more balanced view and include references to much more recent literature clearly showing that signaling of AT1R and V2 does not only require arrestins to engage MAPK signaling from endosomes but occurs in the absence of arrestins and any internalization from the PM as well.

This sentence continues... while β -arrestins mediated ERK1/2 for the B2R, V2R and β -adrenergic receptors (β 1AR and β 2AR) have been shown to also occur at the PM 17-19." Please be more precise and clearly discriminate whether you refer to G protein independent arrestin-mediated ERK or G protein-dependent arrestin-modulated ERK. The term beta arrestin-mediated ERK is semantically imprecise and potentially misleading, particularly as recent literature has shown that arrestins are dispensable for initiation of ERK signaling for both beta2AR and V2R. Reference to this much more recent literature must be made here and the term arrestin-mediated ERK must be avoided unless arrestin's role as transducer or scaffold is clearly stated.

line 76: "receptor/ β -arrestin complex signaling at the PM" please define what this is. which type of signaling do you refer to? G protein-independent signaling or arrestin-contribution to G protein dependent signaling? As there is plenty of misleading literature in this field, please be semantically precise to avoid further confusion.

At the suggestion of the reviewer, we have substantially revised the introduction to provide a more balanced view of the different mechanisms of ERK1/2 activation by GPCR and cited specific examples of receptors, as well as cited recent literature reporting the involvement of G protein and/or β -arrestin's role in MAPK, as assessed using CRISPR-KO cells, including our own work (lines 56-68, 83-99, and 108-112)⁵⁻¹⁶. We have better defined what we mean by G protein- and β -arrestins-mediated MAPK activation, which latter mechanism we now define as β -arrestins-scaffold MAPK signaling. We also better define what type of signaling at the PM we refer to in the manuscript, when appropriate (line 101-104)

Introduction, lines 64-67: "Therefore, the development of new pharmacological tools that block

receptor internalization at distinctive steps would not only provide a unique opportunity for addressing the role of β -arrestins-mediated MAPK signaling in different cell compartments, but to also control specific mitogenic responses in cells." The authors make important claims here in that internalization inhibitors would allow to address the role of arrestins for MAPK signaling.

The reviewer is correct, this is indeed our hypothesis. We now explain in the introduction as why the discovery of new inhibitors acting at different stages of receptor internalization and trafficking would be useful. These could be used in complementary experiments with CRISPR-depleted cells, as suggested by this reviewer and assessed here (see below).

"They identify an inhibitor of internalization and this molecule also inhibits MAPK signaling. But, most importantly, the authors make no effort in this manuscript to show that these processes are intertwined. Thus they MUST examine whether their novel inhibitor rasarfin inhibits MAPK signaling in cells lacking barr1 and barr2, where AT1R does not internalize". If rasarfin loses its effect on MAPK signaling, the links they attempt to make would hold. At present, the link is unsupported. Barbadin, previously published and included in the report, must be used as control under the exact same conditions in barr1/2 depleted cells. both ligands should lose their effect when arrestins are genetically absent. Such data are very simple to generate and of high mechanistic importance to support the above links between trafficking and signaling."

The reviewer is correct that because Rasarfin blocks both internalization and MAPK through respectively ARF6 and Ras inhibition, we cannot determine exactly to what extent these processes are intertwined. We had previously mentioned this caveat in the discussion of the original submitted version and speculated that the development of selective Ras vs. ARF6 analogues of Rasarfin would help address this issue in the future (lines 542-544). We disagree with this reviewer that we previously made no efforts to show that the internalization and signaling are intertwined. Indeed, we reported in the previous version of this study the effects of Ras-DN and ARF-DN on internalization and signaling, respectively (Supplementary Fig. 6). Our results suggest that Ras is most likely not involved in internalization, while for ARF, we observed a tendency to reduce MAPK activation, but the effect was not significant. This is extensively discussed (lines 519-527 and 531-544).

We thank the reviewer for this suggestion to test the effect of Rasarfin on MAPK signalling in β -arrestin1/2 KO cells despite the caveats of using such a model, as previously reported^{6,8}. Indeed, it was shown that compensatory mechanisms that may favor alternative mechanisms for ERK activation occur at different levels in these cells. Nonetheless and as suggested by this reviewer, we assessed the effects of Rasarfin on AT1R-mediated MAPK in β -arrestin1/2 KO cells, which we now present in the new Supplementary Fig. 4a and report the finding in the results section (lines 223-230). We previously reported that despite MAPK activation by AT1R being mainly mediated by G α q/11 at the PM, β -arrestins also contribute to this process via both the regulation of receptor desensitization at the PM and signaling of internalized receptor in endosomes, which correlates with the formation of stable receptor- β -arrestin complexes inside the cells⁵. Therefore, and as expected when β -arrestin1 and 2 are genetically ablated in cells, AT1R-mediated activation of ERK1/2 readily persists (Supplementary Fig. 4a and⁵), because receptors are no longer internalized nor efficiently desensitized for G protein signaling. In such conditions, Rasarfin also totally blocked AT1R-mediated activation of ERK1/2. This, however, doesn't infer

that β -arrestins are not involved in regulating some aspects of MAPK activation, as we have previously shown that genetically altered cells where β -arrestins are removed, cells differentially rewire their signaling path to MAPK⁶. We also assessed Rasarfin's effect in G α q/11 KO cells, where we overexpressed or not β -arrestin to minimize any residual G protein signaling (through desensitization) and potentially favor more β -arrestins-scaffold signalling (Supplementary Fig. 4a). In these condition, Rasarfin also totally blocked AT1R-dependent activation of ERK1/2 in both conditions. These results highlight the difficulty of using only genetically altered cells to characterize which pathways are downstream of GPCRs for MAPK activation and support our premise that pharmacological tools are needed. Notwithstanding such limitation, and as underscored by this reviewer, we also believe that these new results provide mechanistic insights as to how Rasarfin acts as an inhibitor in AT1R-mediated MAPK activation, since they suggest that by engaging either pathway downstream of the receptor to activate MAPK (e.g., G protein-mediated or β -arrestin-scaffold signalling), Ras is a common target. Although the differential roles of G proteins vs. β -arrestins in regulating MAPK is an interesting topic and was the premise for this study to find new molecules that would affect only the latter process, we would also like to emphasise that our study has refocused on the discovery and the characterization of a new Ras and ARF6 inhibitor, which also unveiled new mechanisms of action on Ras.

At the suggestion of this reviewer, we also tested the effect of Barbadin on AT1R-dependent activation of ERK1/2 (see Figure 1, below) and compared it to V2R, which we have previously shown inhibits receptor-mediated internalization (via inhibition of the β -arrestin/AP-2 complexes) and MAPK activation¹⁷. While we still observed potent inhibition of V2R-mediated ERK1/2 phosphorylation in HEK293 cells, Barbadin was ineffective at blocking MAPK promoted by AT1R. While the exact reason for this lack of effect is still unclear, the results may not be that surprising because AT1R no longer internalizes in presence of Barbadin and is therefore trapped at the PM¹⁷ and the canonical G α q/11 pathway for ERK1/2 activation supersedes other mechanisms for MAPK, similar to what we observed in β -arrestin1/2 KO cells⁵. These observations also highlight the complexity in receptor-mediated MAPK signaling mechanisms involved for different GPCRs and again further support the need of tools/inhibitors acting at different steps of receptor internalization for studying the relative role of receptor trafficking in MAPK activation for different GPCRs. We are presently pursuing the characterization of the underlying mechanism for this difference of Barbadin on AT1R-mediated MAPK signaling as well as its effects on different receptors, but we believe that this goes beyond the scope of the present work.

Figure 1: Effect of Barbadin on V2R- and AT1R-mediated MAPK activation. HEK293 cells expressing either V2R or AT1R were pretreated or not (DMSO) with Barbadin (100 μ M), and ERK1/2 activation over time was assessed as described in the manuscript following agonist stimulation. N=3; ****p < 0.0001, ***p < 0.005, two-way ANOVA with Bonferroni correction.

“lines 71-73: AT1R engages MAPK signaling in endosomes... add newer literature to mention that it also signals from the PM in the absence of any internalization. This has clearly been demonstrated but has been omitted so far.”

We have now better explained in the introduction that MAPK signaling can indeed occur from the engagement of receptors at the PM (e.g., without internalization) and cited more contemporaneous literature about that, including our most recent one (see previous answer).

“the selection of compound 21 in Fig1 is too cryptic. Low cell toxicity must be shown for this compound. If cell rounding was analyzed, how does it look like? Why was 21 chosen and not another one that inhibits the BRET signal better? Please consider indicating which of the compounds are toxic at 50 micromolar already in this table in Fig1b. The selection of cpd 21 requires more transparency.”

We apologize if our description of the ligand selection initially sounded cryptic. This was certainly not our intention. We have now added more details about our inclusion/exclusion criteria for compound selection in the results section (lines 150-167). We also better explain why we initially selected **21** over other molecules without dismissing some of the other compounds that could have also been interesting to investigate. We mentioned in our previous version the exclusion of compounds that seemed toxic. However, toxicity was only observed for some compounds that seemingly increased internalization and not for the inhibitors (see examples in Fig. 2 below). Since compounds that increased internalization were not characterized furthermore, we do not think they needed to be highlighted in the Fig. 1b. We have nonetheless modified the text to highlight that none of the inhibitors seemed toxic to cells (lines 154-156).

We also found that some compounds, such as **26**, showed autofluorescence and were thus excluded (Fig. 2, below).

Figure 2: Effects of different compounds on cell morphology. Cells were treated with 50 μ M of compound (or DMSO) and their effects on cell morphology was assessed by microscopy, looking at cells expressing YFP-tagged B2R, which highlights cell contours.

“Results: Figure 1a and Supplementary Table 1. The table is not really helpful as it provides no information to the interested reader. Instead I would request to show the structures of the compounds listed in Fig1b. Only compound 21 is shown, the other active and inactive compounds are also interesting from a structural perspective. Are there analogs with similar profiles? Families of active compounds? All this information is hidden.”

“Same is true for Fig1b and Suppl Table 2, it has no value without structures. Readers must see how selected compounds look like.”

In Fig. 1a, we described the strategy for selecting molecules from our screen, which we have now expended on in the revised results section (also see aforementioned comments). For transparency, we believe it is important to report raw data, hence we included the BRET inhibition data, and GFP and RLucII quench signals from our HTS of the 115,303 compounds we screen (Supplemental Table 1). Selected molecules (Fig. 1b) all had some activity (either inhibiting or increasing receptor targeting to endosomes) but different structures. We appreciate the reviewer’s suggestion to compare active vs. inactive molecules from a structural perspective, to identify classes of molecules, as we did for the series of Compounds **21**. As stated before, all molecules presented in Fig. 1b belong to different sub-ensemble of chemical space, which we now mention in the text as new information (lines 156-159). We are also sensitive to this reviewer’s request of publishing the structures of all compounds, but as he/she can imagine we

are still validating their modes of action; hence we prefer being cautious and not publishing yet their structures before they have all been fully vetted for their mechanisms of action and structure-function relationship. Indeed, the focus of our study was about the discovery of compound **21** and its mode of action and not of the other hundreds of compounds found to have some levels of activity without knowledge about their mechanisms. Nonetheless, we are prepared to provide such information, or other structures from Supplemental Table 1 and 2 upon reasonable request from interested readers. We have now included such a disclaimer in the “Data Availability” section of the manuscript (lines 855-857).

“Fig1e: the concentration of compound 21 must be given in the legend. This is important to allow reproduction of data by others.”

We agree with the reviewer and thank him/her for noting this oversight on our part. We have now provided the concentration of compound 21 (50 μ M) in all figure legends.

“data related to Fig1: please explain why you omitted barr1 in Fig1f. Does cpd 21 inhibit barr1 recruitment to B2R? Was this tested?”

The reviewer must be referring to Suppl. Fig. 1b, since we previously have shown the effect (or lack thereof) of **21** on β -arrestin 1 and 2 recruitment to AT1R (Fig. 1f), but only for β -arrestin2 on B2R (Supplementary Fig. 1b). We now show a similar lack of inhibition of **21** on β -arrestin1 recruitment to B2R, similar to what was reported in the previous version the manuscript with β -arrestin2. We added these new data to the new Supplementary Fig. 1b and modified the Result section accordingly (lines 176-179).

“Fig 2a: what is the concentration of 21? this must be stated. Barbadin is used as reference here at very high molar concentration (100 micromolar). Does Barbadin at this very high concentration induce any toxic effects to cells visible for example in cell rounding? As concentrations are not given for 21 and no statement is made whether the applied concentrations of both compounds are maximally effective, quantitative comparison of arrestin-AP2 complex formation is not permitted. Lacking information must be provided to make sure conclusions are supported by data.”

The concentration of **21** used was 50 μ M, which is close to its maximum solubility concentration in DMSO. We used Barbadin at the highest concentration possible (100 μ M) that also retained solubility in DMSO. At 100 μ M, Barbadin appeared nontoxic to HEK293 cells, since they retained their flat, cobble stone appearance (Fig. 4i)¹⁷, similar to what we observed with **21** (Fig. 1e and g, and Supplementary Fig. 1a of the manuscript, and Fig. 2 of this rebuttal letter). We have now provided such information in the results section and legends. Barbadin at 100 μ M and Rasarfin at 50 μ M represent around 5 and 7 times the IC₅₀ for inhibiting β -arrestins/AP2 interaction (i.e., EC₅₀ of around 20 μ M)¹⁷ and ARF6 (7 μ M, Fig. 4d), respectively. They also have a different mechanism of action, as we now underscore in the discussion section (lines 476-480).

“Are the data shown in Supp Table3 n=1? The number of technical and biological replicates

must be given. If this is a single data point only, it must be stated clearly in the legend as such data are associated with quite some uncertainty.”

As for many HTS, this one done to assess potential off-target effects of Rasarfin was also performed one time, with duplicates. This is now mentioned in the legend of Supplementary Table 3.

Minor Points:

“Abstract: “Internalization and intracellular trafficking of hormone receptors, like receptor tyrosine kinases (RTKs) and G protein-coupled receptors (GPCRs), play pivotal roles in cell responsiveness homeostasis.” PLease check this sentence, is it complete?”

We have corrected the sentence.

Introduction, line 44: "maintain cell responsiveness homeostasis". Is this sentence complete?

“line 176: pathways instead of patways; line 204: also instead of alos”

These typos have been corrected.

Reviewer #2 (Remarks to the Author):

The manuscript "Discovery of dual Ras and ARF6 inhibitor from a GPCR endocytosis screen" by Giubilaro et al. presents results of a HTS, biochemical and in silico work to identify a dual RAS/ARF6 inhibitor. Identification of Rasarfin was based on HTS and validated by several biochemical assays and supported by in silico methods (docking, MD simulations).

I have some very serious concerns over the validity of in silico work. I will analyze here the most critical aspects, without going into the details.

1. Modeling of hRAS: Docking. There is no data (except one image in Figure 5 and one supplementary image) showing the individual docking poses. Especially one would need information if the poses of all compounds do like each other and if individual poses are similar (within the same compound). It is well known that scoring functions do not yield reliable predictions concerning the docking poses, so one must use quite a lot subjective analysis.

We thank this reviewer for his valuable feedback concerning our in-silico work. Changes have been made, and we now believe the manuscript has been significantly improved due to the suggestions of reviewer 2. Docking poses have been visualized in Figure 5a to underline how similar the 10 best scored poses are. Moreover, we present in Supplementary Fig. 7 the single docking poses and the cluster of those poses showing the inverted binding pose. The best ranked pose was chosen, as it shows ideal interaction with the binding groove on Ras, but also mimics SOS interactions, as illustrated in Figure 6c and now mentioned in the manuscript. To furthermore validate our hypothesized binding site, association molecular dynamics were carried

out (Figure 5b and c). Molecular dynamics show how the ligand associates to the suggested binding pocket. Details about these experiments have been specified in the Methods part and in the Results section (lines 313-349, 351-378, 777-781, and 783-813).

“MD simulations: This is a critical issue. All the used simulations are absolutely too short to show any reliable information if hit compounds are binding hRAS or not. Authors should look at two papers how MD simulations should be done and especially how those should be analyzed -(Grossfield A, Patrone P, Roe D, Schultz A, Siderius D, Zuckerman D (2018) Best Practices for Quantification of Uncertainty and Sampling Quality in Molecular Simulations [Article v1.0]. Living Journal of Computational Molecular Science

1: <https://doi.org/10.33011/livecoms.1.1.5067>)

AND

-(Braun E, Gilmer J, Mayes H, Mobley D, Prasad S, Zuckerman D, Monroe J (2018) Best Practices for Foundations in Molecular Simulations [Article v1.0]. Living Journal of Computational Molecular Science 1: . <https://doi.org/10.33011/livecoms.1.1.5957> FOR YOUR INFORMATION, THERE IS NO NEED TO ADD THESE AS A CITATION).

In the current manuscript there is no analysis of the quality of MD (RMSD is not enough here, look the references above!). Also, if you look at the typical timeframe how long simulations are needed for RAS it is clear that at least a microsecond scale is needed. In a kRAS study -(Pantsar T, Rissanen S, Dauch D, Laitinen T, Vattulainen I, Poso A (2018) Assessment of mutation probabilities of KRAS G12 missense mutants and their long-timescale dynamics by atomistic molecular simulations and Markov state modeling. PLoS Computational Biology 14: . <https://doi.org/10.1371/journal.pcbi.1006458>)- simulations were at the level of 5 microsecond/protein. Also, another good example for kRAS MD studies -(Vatansever S, Erman B, Gümüş ZH (2019) Oncogenic G12D mutation alters local conformations and dynamics of K-Ras. Scientific Reports 9:11730 . <https://doi.org/10.1038/s41598-019-48029-z>)- used a microsecond scale simulations.”

“The reason why I also believe these long simulation times are mandatory is coming from the nature of proposed binding mode. The putative binding pocket is in between Switches 1 and 2. Both of the previously mentioned studies found out that exactly these domains are quite flexible . Since the used X-ray structure is based on the SOS1/hRAS complex it has a lot of structural strain due to the existing SOS1 interaction. Once this is removed it is clear that very long simulations are needed before a realistic binding mode can be achieved. Used ns-scale simulations are not more than 1-5% of what is needed. I am aware that this requires a huge computational power, but if this is not done then one cannot claim that MD is indicating anything about the binding.”

We agree with reviewer 2 that the molecular dynamics simulations were relatively short. We therefore extended our simulations reaching an accumulated simulation time of 12 μ s (3 replicas x 4 μ s) for Rasarfin bound to Ras to support our binding hypothesis. Likewise, we extended the simulations for compound **21.4**-bound and the apo forms to 12 μ s (3 replicas x 4 μ s) per system. Altogether, this yielded a total simulation time of 36 μ s. In addition, we addressed the movement of the flexible switches I and II, as suggested by the reviewer (Supplementary Fig. 10). The new Supplementary Figure 10 compares the structural dynamics of switch I and II between the

Rasarfin-bound complex and the apo form of Ras (lines 371-378). Our results highlight how Rasarfin reduces switch flexibility with a potential impact on GTP binding. We also now provide detailed information about the system setups and simulation protocols in the Methods section.

“Pharmacophore analysis should be validated either by searching new compounds OR using some external compounds in the dataset and checking if predictions are working. However, in this case a more robust approach would require simulations of both known active and inactive analogs with hRAS and checking if MD is able to discriminate compounds.”

As the study was conducted starting from an HTS, we did not consider searching for more analogues using the pharmacophore model. We chose Rasarfin based on biological data and selected structurally similar analogues to discriminate between molecule inhibitors vs. non-inhibitors. We used the pharmacophoric approach to deduce essential pharmacophoric features on the small molecules, which are responsible for crucial interactions and binding to the protein interface. Regarding the simulation of the active vs. the inactive analogs, it is what we are now showing in the MD approach by comparing Rasarfin possessing all relevant pharmacophoric features, to compound **21.4**, which is lacking a bulky, hydrophobic feature (the chloro atom), which turns out to be crucial for binding. We consider that further screening is out of the scope of our work. Furthermore, we provided the biological support for the essential feature, which discriminates between active and inactive compounds through mutational work (Supplementary Fig. 11).

“Why not ARF6 modeling at all? It is quite probably that binding mode is similar between hRAS and ARF6.”

We agree with this reviewer that defining the binding mode of Rasarfin onto ARF6 could be interesting. However, due to the lack of a crystal structure of ARF6, we cannot reliably determine binding modes. Modeling work would have to be based on a homology model and rely on assumptions about the effector binding, which involves a lot of protein flexibility and rearrangement, which would eventually lead to more questions and uncertainties. This aspect cannot be reliably addressed with modelling work only. Therefore, at this point, we think it is out of the scope of our paper to provide these speculative results. Such limitations were discussed in the text in the previous version of the manuscript (lines 480-488).

A minor comment: please check the simulation videos (supplementary material), at least one simulation includes extra material (menus down right corner) and a suspicious jump in the middle of trajectory.

We thank the reviewer for the comment. The videos have been updated using the trajectories of the new production runs and made available in the Supplementary Material as two new videos (Supplementary Video 1 and 2).

- 1 Raimbourg, J. *et al.* Sensitization of EGFR Wild-Type Non-Small Cell Lung Cancer Cells to EGFR-Tyrosine Kinase Inhibitor Erlotinib. *Mol Cancer Ther* **16**, 1634-1644, doi:10.1158/1535-7163.MCT-17-0075 (2017).
- 2 Arakaki, A. K. S., Pan, W. A. & Trejo, J. GPCRs in Cancer: Protease-Activated Receptors, Endocytic Adaptors and Signaling. *Int J Mol Sci* **19**, doi:10.3390/ijms19071886 (2018).
- 3 Laporte, S. A. & Scott, M. G. H. beta-Arrestins: Multitask Scaffolds Orchestrating the Where and When in Cell Signalling. *Methods Mol Biol* **1957**, 9-55, doi:10.1007/978-1-4939-9158-7_2 (2019).
- 4 Peterson, Y. K. & Luttrell, L. M. The Diverse Roles of Arrestin Scaffolds in G Protein-Coupled Receptor Signaling. *Pharmacol Rev* **69**, 256-297, doi:10.1124/pr.116.013367 (2017).
- 5 Cao, Y. *et al.* Angiotensin II type 1 receptor variants alter endosomal receptor-beta-arrestin complex stability and MAPK activation. *J Biol Chem* **295**, 13169-13180, doi:10.1074/jbc.RA120.014330 (2020).
- 6 Luttrell, L. M. *et al.* Manifold roles of beta-arrestins in GPCR signaling elucidated with siRNA and CRISPR/Cas9. *Sci Signal* **11**, doi:10.1126/scisignal.aat7650 (2018).
- 7 Jain, R., Watson, U., Vasudevan, L. & Saini, D. K. ERK Activation Pathways Downstream of GPCRs. *Int Rev Cell Mol Biol* **338**, 79-109, doi:10.1016/bs.ircmb.2018.02.003 (2018).
- 8 Gurevich, V. V. & Gurevich, E. V. Arrestins and G proteins in cellular signaling: The coin has two sides. *Sci Signal* **11**, doi:10.1126/scisignal.aav1646 (2018).
- 9 Grundmann, M. & Kostenis, E. Temporal Bias: Time-Encoded Dynamic GPCR Signaling. *Trends Pharmacol Sci* **38**, 1110-1124, doi:10.1016/j.tips.2017.09.004 (2017).
- 10 Grundmann, M. *et al.* Lack of beta-arrestin signaling in the absence of active G proteins. *Nat Commun* **9**, 341, doi:10.1038/s41467-017-02661-3 (2018).
- 11 O'Hayre, M. *et al.* Genetic evidence that beta-arrestins are dispensable for the initiation of beta2-adrenergic receptor signaling to ERK. *Sci Signal* **10**, doi:10.1126/scisignal.aal3395 (2017).
- 12 Houry, E., Nikolajev, L., Simaan, M., Namkung, Y. & Laporte, S. A. Differential regulation of endosomal GPCR/beta-arrestin complexes and trafficking by MAPK. *J Biol Chem* **289**, 23302-23317, doi:10.1074/jbc.M114.568147 (2014).
- 13 Paradis, J. S. *et al.* Receptor sequestration in response to beta-arrestin-2 phosphorylation by ERK1/2 governs steady-state levels of GPCR cell-surface expression. *Proc Natl Acad Sci U S A* **112**, E5160-5168, doi:10.1073/pnas.1508836112 (2015).
- 14 Namkung, Y. *et al.* Functional selectivity profiling of the angiotensin II type 1 receptor using pathway-wide BRET signaling sensors. *Sci Signal* **11**, doi:10.1126/scisignal.aat1631 (2018).
- 15 Schrage, R. *et al.* The experimental power of FR900359 to study Gq-regulated biological processes. *Nat Commun* **6**, 10156, doi:10.1038/ncomms10156 (2015).
- 16 Zimmerman, B. *et al.* Role of ssarrestins in bradykinin B2 receptor-mediated signalling. *Cell Signal* **23**, 648-659, doi:10.1016/j.cellsig.2010.11.016 (2011).

- 17 Beaufrait, A. *et al.* A new inhibitor of the beta-arrestin/AP2 endocytic complex reveals interplay between GPCR internalization and signalling. *Nat Commun* **8**, 15054, doi:10.1038/ncomms15054 (2017).

Reviewers' Comments:

Reviewer #1:

Remarks to the Author:

This reviewer appreciates the efforts of the authors to strengthen their interesting manuscript further by writing and experimentation as well as elegant additional computational studies. Overall, there is much improvement after the revision but there are still a number of issues that remain to be improved and clarified. I also feel that abstract and discussion fit well, but the intro is somewhat disconnected from abstract and discussion. Abstract and discussion heavily focus on cancer but the Intro largely focusses on signaling mechanism which have prevailed in the field for years/decades but are, unfortunately, conceptually misleading. Therefore, these long held tenets must be treated with more care. While I do appreciate the authors citing original literature and I very much agree that this should be done, I suggest and request that, once we know better, should not insist on old concepts, particularly if these have never been truly supported by experimental data. Now, new data, new tools and technologies are in place, and require cognitive flexibility to re-interpret some of the older claims and to reconsider whether these long lasting tenets still hold. Particularly the Introduction must acknowledge development of the field to a larger extent including citation of appropriate references as will be specified below.

Specific comments:

Abstract

1. "play pivotal roles in cell responsiveness homeostasis." Is this sentence correct and complete? Please correct.

Introduction

1. lines 75-77 " β -arrestins have also been shown to act as signaling adaptors by recruiting signaling proteins, notably elements of the MAPK cascade such as Raf-1, MEK and ERK1/2 itself." Please support this essential statement with an appropriate citation. Which paper has shown this recruitment of signaling proteins? It is of high relevance to readers to be referred to quintessential literature, the basis for the arrestin signaling scaffold concept. No review must be cited here, the original report is required. Of course individual supporting manuscripts may all be cited.

2. lines 82-84: " β -arrestin-scaffold MAPK signaling for GPCRs also likely involves, in some instances, some levels of G protein activation and/or RTK transactivation^{9,23}." Please rephrase as this does not reflect the current knowledge in mechanisms of MAPK signaling, for example " β -arrestin-scaffold MAPK signaling for GPCRs also involves G protein-driven initiation and/or RTK transactivation^{9,23}." The authors need to make sure that G protein contribution to this signaling event is adequately reflected. It here reads as a rather vague option, however a number of solid and well-controlled papers attest to G protein-driven initiation. The term "some levels of G protein activation" is clearly inappropriate. Reference 9 must not be cited, instead Alvarez-Curto, JBC2016, O'Hayre Science Signaling 2017 and Grundmann et al, Nature Communications 2018 must be cited here to reflect that there is no MAPK signals if G proteins are not activated.

3. lines 85/86 "Despite that GPCRs engage MAPK through both G protein-dependent and β -arrestin scaffold mechanisms," This reads as if there are 2 distinct mechanisms but there is no evidence for a β -arrestin scaffold mechanism if G proteins haven't triggered the process. MAPK signaling is G protein initiated and may be β -arrestin scaffolded but there is no evidence for an exclusively β -arrestin scaffolded process. All studies claiming β -arrestin signaling somewhat ignore/overlook the presence of G proteins. However, G protein presence and activation is relevant as they initiate the signal which is then modulated by an arrestin scaffolded mechanism. This must be acknowledged semantically by the authors to avoid conveying the existence of 2 mechanisms, for which there is no supporting

evidence. The signaling process they refer to is a G protein-initiated beta arrestin scaffolded one, just as published in Trefier et al., (G protein-dependent signaling triggers a β -arrestin-scaffolded p70S6K/rpS6 module that controls 5'TOP mRNA translation, FASEB Journal 2018) for another biological process controlled in a comparable way by G proteins and arrestins.

4.lines 85-88: "Despite that GPCRs engage MAPK through both G protein-dependent and β -arrestin scaffold mechanisms, the relative contribution of these two classes of effectors in each other's signaling pathway and in the overall mitogenic cell response remains an open question, and the subject of recent investigation." It only remains an open question as the field somehow ignores most relevant recent literature that offers clarification of apparently enigmatic mechanisms. Again, please rephrase to not imply 2 distinct mechanisms, you can only discuss relative contribution of arrestins but they only contribute to a G protein initiated signaling event. It is incorrect to imply existence of 2 mechanisms, which is frequently done in high impact papers but experimentally unsupported. There is no MAPK signaling if G proteins are inactive; numerous old papers attest to this as well (no pERK for Gi GPCRs when PTX pretreated), you may cite these numerous old papers in addition to the newer ones using genome-edited cells and mentioned above.

5. 96-98 "We indeed observed for AT1R that ERK1/2 signaling was increased in β -arrestin1/2 KO HEK293 cells compared to parental cells, consistent with the lack of receptor desensitization, internalization and increased G protein signaling at the PM15". While ref 15 shows this, Grundmann et al NCOMMS 2018 shows exactly the opposite and other older literature as well (using arrestin knockdown approaches). Therefore, citing half of the truth shouldn't be used in support of 2 distinct mechanisms, data only show that deletion of arrestins affects pERK to different extents, and may enhance or decrease pERK for AT1, exactly as expected for one mechanism that is G protein-driven and arrestin scaffolded.

6.lines 99-101. This reviewer disagrees with the interpretation of the observed rewiring. No matter if arrestins are deleted completely or partially, pERK remains detectable in all instances. The observed differences in ref21 are in part statistically significant but not biologically convincing to support any rewiring theories. Ref 21 data are identical to those in O'Hayre et al., Science Signaling where experimental data entirely support the claims. What is supported in ref 21 is a G protein-initiated arrestin scaffolded process that explains all data and is in perfect agreement with the three studies listed above. Ref 21 has somehow "forgotten" to take G proteins into account that are the reason for the somewhat variable pERK signals, variability of which is explained by assuming arrestins acting as simple scaffolds that bring proteins together but don't signal on their own.

7. Overall, please make sure that the Intro is freed from distinguishing two mechanisms for which there is no experimental evidence (only numerous unsupported claims). Please also make sure that the Intro clearly conveys that MAPK signaling does not require arrestins, occurs in their genetic absence (both knockout and knockdown), does not occur when chemical inhibitors for G proteins are used (like PTX, YM,FR, so in the absence of any "rewiring"), yet may be modulated to variable degrees by their presence. To corroborate your statements, cite literature on arrestin biased receptors which have in common that they recruit arrestins but don't signal.

8. another relevant and general statement: The authors are worried about caveats of using genetically altered cells, rewiring, moving MAPK signaling towards G protein-dependent routes. I don't think that the authors need to worry that much because all papers examining pERK of Gi-GPCRs show no pERK after PTX pretreatment clearly showing the G protein relevance. If they wish to keep the rewiring theory, they must discuss G protein contribution in a more balanced way and mention all the references in which pERK is extinguished when G proteins are inhibited pharmacologically clearly showing G protein roles without any rewiring.

10. Please state clearly (and provide suitable references) whether β -arrestin-scaffolded MAPK signaling from endosomes requires prior activation of G proteins or whether this mechanism is truly G

protein-independent arrestin scaffolded. This is to clarify the statement in lines 104-106.

11. Overall, the Introduction requires more clarity and a more balanced view of currently existing and experimentally supported signaling theories. Not least to better support the conceptual value of a dual ras arf inhibitor. The value of specific inhibitors is clear, a dual inhibitor is an interesting molecule, but why should we care about inhibition of ARF6 when we inhibit Ras? Inhibition of ARF6 would be dispensable at least in theory. These are thoughts addressed in the discussion but they are already relevant in the Introduction.

Results

1. lines 157-158: please re-read the sentence. It is not clear. It implies that the inhibitors are treated with compounds rather than the cells.

2. Fig1f: a heading would help to know AngII-promoted BRET of which partners is measured. Indicate receptor and label like in panel 1e or g. This enhances clarity.

3. Fig1d: Indicate the BRET assay window somewhere (legend for example) so that readers know the size of the inhibited assay window.

4. lines 227-234, newly added text; please rephrase for clarity and without the assumption of an arrestin scaffold pathway that functions in isolation. arrestin-scaffolded MAPK phosphorylation is G protein-dependent. The current description of the results does not benefit from a distinction between G protein dependent and arrestin scaffold signaling (please note that G protein signaling may be arrestin scaffolded and arrestin scaffold signaling requires active G proteins which means it is one and the same process but not two. Instead of "to favor more internalized receptor and β -arrestin-scaffold MAPK signaling." you may state to favor more internalized receptors...Clearly 21 inhibition of ERK phosphorylation does not depend on Gq/11 signaling and occurs with endogenous and overexpressed arrestins. In the opinion of this reviewer 21 inhibition pERK works best when arrestins are gone. Why? I urge and invite the authors to speculate about this apparent correlation. Could you imagine a reason why the absence of arrestins would favor the ERK inhibitory effect of 21 while the presence of arrestins might tend to diminish it? There seems to be a correlation. What could it mean? In other words, to me it seems as if 21 works best when internalization is blocked and effects are diminished when internalization is favored. One last thought: is it possible that arrestins by themselves diminish access of 21 to Ras because of their scaffold function?

5. lines 308,309:simulation. "ARF6-T27N did not have a significant effect on AngII-mediated ERK1/2 stimulation, while K-Ras-S17N efficiently inhibited the mitogenic response (Supplementary Fig. 6b and c). Looking at SF6 implies a slight trend towards more pERK in the presence of the dn-ARF (not statistically significant). This would agree well with data in Fig2d,e, showing less AT1R internalization in the presence of dn-ARF6. Together this would suggest that internalized AT1R is not the relevant species for MAPK signaling but rather PM-AT1R. This would be entirely consistent with findings of the Shukla lab, published in 2017 in Nature Nanotechnology for numerous receptors (AT1 not included though). It would be another reason to defocus from endosomal G protein-independent arrestin scaffold signaling, first because the process as such is not sufficiently supported and second because data in this very paper would rather imply the opposite. Regardless, the dubious value of an unsupported signaling mechanism should not diminish the value of 21 and its capacity to inhibit internalization and MAPK signaling.

6. If 21 interferes with the ability of SOS to promote guanine nucleotide exchange on Ras, could the presence of arrestins impair 21 inhibition of the SOS-Ras interaction? Is the Ras groove, identified as the preferred binding site of Rasarfin, less accessible to 21 when arrestins are present? This could directly link to the speculation I suggested above and could be analyzed computationally by the newly involved coauthors. It would also add to the elegant computational work that has already been added

to this paper.

7. Discussion, 546-549 "On the other hand, Rasarfin, which also inhibited receptor internalization, efficiently blocked receptor-mediated ERK1/2 activation. Such effect, however, cannot be totally attributed to the inhibition of receptor internalization by Rasarfin since it also acted on Ras." I think the author's own data suggest that Rasarfin inhibition of pERK is not related to inhibition of internalization, just the opposite seems to be the case (see my comment from above and compare results in Fig 2d,e and Suppl Fig6b). Please make sure you discuss the contribution of internalized AT1R to pERK in a more balanced way to not contradict your own newly presented data. Instead, you may cite more literature clearly showing that inhibition of internalization of GPCRs rather enhances pERK (Shukla, Nat Nanotech for example).

8. Overall, looking at the entire paper again, there is much improvement after the revision but much remains to be improved and clarified based on the new additions.

Reviewer #2:

Remarks to the Author:

Initially I had two main issues with this manuscript, MD simulations and quality of docking results (the latter was more about reporting issue than actually how docking was carried out).

These issues are now solved.

REVIEWER COMMENTS AND ANSWERS

We thank both reviewers for their careful re-assessment of our work and their constructive comments. We are happy that reviewer 2 found that our revised manuscript addressed all the concerns previously raised. Below are our answers to the reviewers' questions/comments. Modifications to the text can also be found in the appended track-change version of our manuscript.

Reviewer #1 (Remarks to the Author):

This reviewer appreciates the efforts of the authors to strengthen their interesting manuscript further by writing and experimentation as well as elegant additional computational studies. Overall, there is much improvement after the revision but there are still a number of issues that remain to be improved and clarified. I also feel that abstract and discussion fit well, but the intro is somewhat disconnected from abstract and discussion. Abstract and discussion heavily focus on cancer but the Intro largely focusses on signaling mechanism which have prevailed in the field for years/decades but are, unfortunately, conceptually misleading. Therefore, these long held tenets must be treated with more care. While I do appreciate the authors citing original literature and I very much agree that this should be done, I suggest and request that, once we know better, should not insist on old concepts, particularly if these have never been truly supported by experimental data. Now, new data, new tools and technologies are in place, and require cognitive flexibility to re-interpret some of the older claims and to reconsider whether these long lasting tenets still hold. Particularly the Introduction must acknowledge development of the field to a larger extent including citation of appropriate references as will be specified below.

We thank this reviewer for his/her positive comments about our efforts to strengthen our data and our new computational data. We have better streamlined our introduction and now present what we believe is a balanced overview of the mechanism leading to MAPK activation by GPCRs. Importantly we have made it clearer in the introduction that β -arrestin involvement in MAPK doesn't exclude G protein activation, hence they are not necessarily two independent mechanisms. We have also downplayed in our rationale the role of endosomal- β -arrestin mediated MAPK. We nonetheless believe it is important to summarize some of the mechanisms and supporting evidence about the different mechanism for ERK1/2 activation by GPCR, since our new inhibitor (Rasarfin) is blocking such response, and because we present data using G protein and β -arrestin KO cells (as previously requested by this reviewer), and because there is ample reported evidence in the literature describing the involvement of both G proteins and β -arrestin in regulating MAPK by GPCRs. We have cited specific studies in our manuscript (see below) about new developments in the field regarding the role of G proteins and β -arrestin in different aspects of MAPK activation by GPCR including many reviews on the subject (Peterson et al., 2017; Grundmann et al., 2017; Luttrell et al., 2003; Lefkowitz, & Shenoy, 2005, and Gurevich, & Gurevich, 2018). Importantly, we are now clearly acknowledging the reported role of G proteins in β -arrestin-mediated MAPK signaling by GPCR, citing studies by O'Hayre et al., 2017; Alvarez-Curto et al., 2016 and Grundmann et al., 2018, as requested.

Specific comments:

Abstract

1. "play pivotal roles in cell responsiveness homeostasis." Is this sentence correct and complete? Please correct.

We have changed the sentence to “...play pivotal roles in cell responsiveness”

Introduction

1. lines 75-77 " β -arrestins have also been shown to act as signaling adaptors by recruiting signaling proteins, notably elements of the MAPK cascade such as Raf-1, MEK and ERK1/2 itself." Please support this essential statement with an appropriate citation. Which paper has shown this recruitment of signaling proteins? It is of high relevance to readers to be referred to quintessential literature, the basis for the arrestin signaling scaffold concept. No review must be cited here, the original report is required. Of course individual supporting manuscripts may all be cited.

We have cited original papers showing the adaptor roles of β -arrestins in signaling (McDonald et al. 2000, and Luttrell et al., 2001). Despite many other studies having confirmed these findings, we are limiting ourselves to citing only those two papers because many reviews that we are referencing in the manuscript are also citing these other studies.

" β -arrestins have also been shown to act as signaling adaptors by recruiting signaling proteins, notably kinases of the MAPK cascade such as Raf-1, MEK and ERK1/2 itself (McDonald, P. et al. 2000 and Luttrell, LM et al. 2001)."

2. lines 82-84: " β -arrestin-scaffold MAPK signaling for GPCRs also likely involves, in some instances, some levels of G protein activation and/or RTK transactivation^{9,23}." Please rephrase as this does not reflect the current knowledge in mechanisms of MAPK signaling, for example "However, β -arrestin-scaffold MAPK signaling for GPCRs also involves G protein-driven initiation and/or RTK transactivation^{9,23}." The authors need to make sure that G protein contribution to this signaling event is adequately reflected. It here reads as a rather vague option, however a number of solid and well-controlled papers attest to G protein-driven initiation. The term "some levels of G protein activation" is clearly inappropriate. Reference 9 must not be cited, instead Alvarez-Curto, JBC2016, O'Hayre Science Signaling 2017 and Grundmann et al, Nature Communications 2018 must be cited here to reflect that there is no MAPK signals if G proteins are not activated.

We have made the appropriate changes in the text to reflect that β -arrestin-mediated MAPK (e.g., ERK1/2) does not necessary imply the lack of G protein signaling in that process and that those 2 mechanisms are not necessarily independent.

“However, because β -arrestins themselves do not possess enzymatic activity, MAPK like Raf-1 must be activated by other mechanisms, such as by the activation G proteins or RTK transactivation²⁰. Indeed, recent studies have reported the absence of ERK1/2 activation by GPCRs in cells lacking G protein activity, hence suggesting a role of G protein-mediated signaling in supporting MAPK signaling by β -arrestins (it has to be understood hereafter that reference to β -arrestin-dependent MAPK signaling does not imply the lack of involvement of G protein signaling in such process (O’Hayre, M et al., 2017; Alvarez-Curto et al., 2016 and Grundmann, M et al., 2018) (it has to be understood hereafter that reference to β -arrestin-dependent MAPK signaling does not imply the lack of involvement of G protein signaling in such process)”.

The O’Hayre et al., 2017; Alvarez-Curto et al., 2016 and Grundmann et al., 2018 studies are now clearly referred to as stand-alone citations in the text to support the proposed roles of G proteins in MAPK signaling by β -arrestins. The former citation 9 (review by Grundmann. and Kostenis, 2017) was removed and the text only now mentions the Gurevich, & Gurevich, 2018 review, which also provides a nice balanced view of the role of G proteins and β -arrestins in MAPK signaling processes.

3.lines 85/86 "Despite that GPCRs engage MAPK through both G protein-dependent and β -arrestin scaffold mechanisms," This reads as if there are 2 distinct mechanisms but there is no evidence for a β -arrestin scaffold mechanism if G proteins haven't triggered the process. MAPK signaling is G protein initiated and may be β -arrestin scaffolded but there is no evidence for an exclusively β arrestin scaffolded process. All studies claiming β arrestin signaling somewhat ignore/overlook the presence of G proteins. However, G protein presence and activation is relevant as they initiate the signal which is then modulated by an arrestin scaffolded mechanism. This must be acknowledged semantically by the authors to avoid conveying the existence of 2 mechanisms, for which there is no supporting evidence. The signaling process they refer to is a G protein-initiated β arrestin scaffolded one, just as published in Trefier et al., (G protein-dependent signaling triggers a β -arrestin-scaffolded p70S6K/ rpS6 module that controls 5'TOP mRNA translation, FASEB Journal 2018) for another biological process controlled in a comparable way by G proteins and arrestins.

We have revised part of the introduction and made it clearer that G protein-dependent and β -arrestin-dependent MAPK are not necessarily independent since several papers reported that G proteins are needed for the β -arrestin-mediated regulation of MAPK. Please see our previous answer to comment #2.

4.lines 85-88: "Despite that GPCRs engage MAPK through both G protein-dependent and β -arrestin scaffold mechanisms, the relative contribution of these two classes of effectors in each other’s signaling pathway and in the overall mitogenic cell response remains an open question, and the subject of recent investigation." It only remains an open question as the field somehow ignores most relevant recent literature that offers clarification of apparently enigmatic mechanisms. Again, please rephrase to not imply 2 distinct mechanisms, you can only discuss relative contribution of arrestins but they only contribute to a G protein initiated signaling event. It is incorrect to imply existence of 2 mechanisms, which is frequently done in high impact

papers but experimentally unsupported. There is no MAPK signaling if G proteins are inactive; numerous old papers attest to this as well (no pERK for Gi GPCRs when PTX pretreated), you may cite these numerous old papers in addition to the newer ones using genome-edited cells and mentioned above.

We have removed the text alluding to the open question relative to the contribution of G protein and β -arrestin in MAPK signaling. We made it clearer not to infer (by omission) that G protein-mediated and β -arrestin-mediated MAPK are independent of each other and cited the recent papers using genome-edited cells mentioned above (please refer to our previous answers #2). We do not feel it is necessary to expend on that topic by citing older papers since it is covered in the many reviews we are referencing to in our manuscript.

5. 96-98 "We indeed observed for AT1R that ERK1/2 signaling was increased in β -arrestin1/2 KO HEK293 cells compared to parental cells, consistent with the lack of receptor desensitization, internalization and increased G protein signaling at the PM15". While ref 15 shows this, Grundmann et al NCOMMS 2018 shows exactly the opposite and other older literature as well (using arrestin knockdown approaches). Therefore, citing half of the truth shouldn't be used in support of 2 distinct mechanisms, data only show that deletion of arrestins affects pERK to different extents, and may enhance or decrease pERK for AT1, exactly as expected for one mechanism that is G protein-driven and arrestin scaffolded.

We have separated these 2 citations and made sure to properly cite the Grundmann et al. study reporting that the lack of G protein activity leads to the absence of ERK1/2 activation by GPCRs, so that there is no confusion with the Cao, Y. et al. 2020 study (previously reference 15).

6.lines 99-101. This reviewer disagrees with the interpretation of the observed rewiring. No matter if arrestins are deleted completely or partially, pERK remains detectable in all instances. The observed differences in ref21 are in part statistically significant but not biologically convincing to support any rewiring theories. Ref 21 data are identical to those in O'Hayre et al., Science Signaling where experimental data entirely support the claims. What is supported in ref 21 is a G protein-initiated arrestin scaffolded process that explains all data and is in perfect agreement with the three studies listed above. Ref 21 has somehow "forgotten" to take G proteins into account that are the reason for the somewhat variable pERK signals, variability of which is explained by assuming arrestins acting as simple scaffolds that bring proteins together but don't signal on their own.

We have removed this sentence alluding to "rewiring" because we believe it is not relevant to the understanding of our findings, nor to the rationale of the study.

7. Overall, please make sure that the Intro is freed from distinguishing two mechanisms for which there is no experimental evidence (only numerous unsupported claims). Please also make sure that the Intro clearly conveys that MAPK signaling does not require arrestins, occurs in their genetic absence (both knockout and knockdown), does not occur when chemical inhibitors for G proteins are used (like PTX, YM,FR, so in the absence of any "rewiring"), yet may be modulated to variable degrees by their presence. To corroborate your statements, cite literature on arrestin biased receptors which have in common that they recruit arrestins but don't signal.

As stated in our answer #2, we have rewritten part of the introduction and made it clearer that G protein-dependent and β -arrestin-dependent MAPK are not necessarily independent processes. We have cited the studies by Grundmann et al. and O'Hayre et al. that support that they are not independent mechanisms. We do not feel that there is a need to expand on that concept, by citing literature on β -arrestin biased receptors, since this is not the focus of our study, and we are already referencing many other reviews on the topic.

8. another relevant and general statement: The authors are worried about caveats of using genetically altered cells, rewiring, moving MAPK signaling towards G protein-dependent routes. I don't think that the authors need to worry that much because all papers examining pERK of Gi-GPCRs show no pERK after PTX pretreatment clearly showing the G protein relevance. If they wish to keep the rewiring theory, they must discuss G protein contribution in a more balanced way and mention all the references in which pERK is extinguished when G proteins are inhibited pharmacologically clearly showing G protein roles without any rewiring.

We have removed this section about the caveats of using genetically altered cells and rewiring in the introduction since it not necessary for the rationale of the study and for the understanding of our findings.

10. Please state clearly (and provide suitable references) whether β -arrestin-scaffolded MAPK signaling from endosomes requires prior activation of G proteins or whether this mechanism is truly G protein-independent arrestin scaffolded. This is to clarify the statement in lines 104-106.

This sentence was removed because it is not necessary for the rationale of the study and for understanding our findings.

11. Overall, the Introduction requires more clarity and a more balanced view of currently existing and experimentally supported signaling theories. Not least to better support the conceptual value of a dual ras arf inhibitor. The value of specific inhibitors is clear, a dual inhibitor is an interesting molecule, but why should we care about inhibition of ARF6 when we inhibit Ras? Inhibition of ARF6 would be dispensable at least in theory. These are thoughts addressed in the discussion but they are already relevant in the Introduction.

We have better introduced the role of ARF6 in regulating receptor internalization in the introduction, which we now believe contextualizes our findings and rationale about Rasarfin working as an inhibitor of GPCR internalization, and why developing inhibitors of internalization and/or MAPK signaling would be useful for regulating GPCR responsiveness. This is also taken in the context, underscored in the introduction, that MAPKs have been shown to regulate receptor trafficking.

Here are some of the added statements:

“Moreover, β -arrestins bind the small GTPase Arf6, which also aid in recruiting AP-2 and clathrin to regulate receptor internalization (Claing, A. et al. 2001 and Poupart et al., 2007)”

“The development of selective inhibitors of receptor internalization/trafficking and/or signaling would therefore allow us to better understand how each of these cell processes regulate the overall GPCR responsiveness, and to what extent they are intertwined”

Results

1. lines 157-158: please re-read the sentence. It is not clear. It implies that the inhibitors are treated with compounds rather than the cells.

Thanks for noting this lack of clarity of the sentence. The sentence has now been re-written to: *“None of the inhibitors tested caused abnormal cell morphological changes when cells were treated with compounds as compared to DMSO”* for more clarity.

2. Fig1f: a heading would help to know AngII-promoted BRET of which partners is measured. Indicate receptor and label like in panel 1e or g. This enhances clarity.

A heading was added to the figure to indicate the BRET partners (e.g., AT1R-YFP/ β arr-RLucII).

3. Fig1d: Indicate the BRET assay window somewhere (legend for example) so that readers know the size of the inhibited assay window.

The window response was added to the Fig 1's legend. It now reads:

“The mean values of the ligand-promoted BRET responses ($BRET_{ligand} - BRET_{basal}$) in DMSO were 0.260, 0.549, and 1.061 for AT1R, B2R, and β 2AR, respectively.”

4. lines 227-234, newly added text; please rephrase for clarity and without the assumption of an arrestin scaffold pathway that functions in isolation. arrestin-scaffolded MAPK phosphorylation is G protein-dependent. The current description of the results does not benefit from a distinction between G protein dependent and arrestin scaffold signaling (please note that G protein signaling may be arrestin scaffolded and arrestin scaffold signaling requires active G proteins which means it is one and the same process but not two. Instead of "to favor more internalized receptor and β -arrestin–scaffold MAPK signaling." you may state to favor more internalized receptors... Clearly 21 inhibition of ERK phosphorylation does not depend on Gq/11 signaling and occurs with endogenous and overexpressed arrestins. In the opinion of this reviewer 21 inhibition pERK works best when arrestins are gone. Why? I urge and invite the authors to speculate about this apparent correlation. Could you imagine a reason why the absence of arrestins would favor the ERK inhibitory effect of 21 while the presence of arrestins might tend to diminish it? There seems to be a correlation. What could it mean? In other words, to me it seems as if 21 works best when internalization is blocked and effects are diminished when internalization is favored. One last thought: is it possible that arrestins by themselves diminish access of 21 to Ras because of their scaffold function?

We have revised the sentence to: *“21 efficiently inhibited ERK1/2 activation by AT1R in cells lacking $G_{\alpha q/11}$ and either supplemented or not with β -arrestin, which favor more receptor*

internalization, hence potentially limiting G protein–dependent signaling at the PM and favoring more β -arrestin– scaffold MAPK signaling”.

We do not agree with this reviewer’s interpretation of our data that **21** is more efficacious at inhibiting MAPK when β -arrestins are absent (e.g., when internalization is prevented by the absence of β -arrestins, as suggested by this reviewer) (Supplementary Fig 4a). We observed total inhibition of AngII-mediated ERK1/2 activation (e.g., not significantly different from basal) in cells treated with **21** when β -arrestins were absent, present or over-expressed in cells (Supplementary Fig 4a). We do not believe that there is any basis to speculate about the potential role of β -arrestins on **21**’s inhibitory effect in MAPK activation by AT1R from what seems nonsignificant differences between these conditions.

5. lines 308,309:simulation. "ARF6-T27N did not have a significant effect on AngII-mediated ERK1/2 stimulation, while K-Ras-S17N efficiently inhibited the mitogenic response (Supplementary Fig. 6b and c). Looking at SF6 implies a slight trend towards more pERK in the presence of the dn-ARF (not statistically significant). This would agree well with data in Fig2d,e, showing less AT1R internalization in the presence of dn-ARF6. Together this would suggest that internalized AT1R is not the relevant species for MAPK signaling but rather PM-AT1R. This would be entirely consistent with findings of the Shukla lab, published in 2017 in Nature Nanotechnology for numerous receptors (AT1 not included though). It would be another reason to defocus from endosomal G protein-independent arrestin scaffold signaling, first because the process as such is not sufficiently supported and second because data in this very paper would rather imply the opposite. Regardless, the dubious value of an unsupported signaling mechanism should not diminish the value of **21** and its capacity to inhibit internalization and MAPK signaling.

We have made it clearer in the text that MAPK signaling, which can involve the adaptor function of β -arrestin is not necessarily independent of G protein activity (please also refer to our previous answers). We also have deemphasized the notion of endosomal β -arrestins scaffold signaling in the introduction and in our rationale (please also refer to our previous answer #11).

6. If **21** interferes with the ability of SOS to promote guanine nucleotide exchange on Ras, could the presence of arrestins impair **21** inhibition of the SOS-Ras interaction? Is the Ras groove, identified as the preferred binding site of Rasarfin, less accessible to **21** when arrestins are present? This could directly link to the speculation I suggested above and could be analyzed computationally by the newly involved coauthors. It would also add to the elegant computational work that has already been added to this paper.

We thank the reviewer for this suggestion. However, in order to perform meaningful MD simulation, one must start with existing structural information about protein and/or ligands complexes. In that respect there is no evidence thus far to suggest that β -arrestin binds directly to Ras or SOS. To our knowledge, no crystal structures of Ras or SOS in complex with β -arrestin have been reported. Such structural knowledge would be required to performed MD simulation to model the effect of β -arrestin on **21**’s binding to Ras.

7. Discussion, 546-549 "On the other hand, Rasarfin, which also inhibited receptor internalization, efficiently blocked receptor-mediated ERK1/2 activation. Such effect, however, cannot be totally attributed to the inhibition of receptor internalization by Rasarfin since it also acted on Ras." I think the author's own data suggest that Rasarfin inhibition of pERK is not related to inhibition of internalization, just the opposite seems to be the case (see my comment from above and compare results in Fig 2d,e and Suppl Fig6b). Please make sure you discuss the contribution of internalized AT1R to pERK in a more balanced way to not contradict your own newly presented data. Instead, you may cite more literature clearly showing that inhibition of internalization of GPCRs rather enhances pERK (Shukla, Nat Nanotech for example).

We agree with this reviewer and have changed the sentence to: *“Such latter effect, however, cannot be attributed to the inhibition of receptor internalization by Rasarfin since it also potently inhibited Ras”*.

We have cited the Gosh et al. 2017 paper (Shukla, Nat Nanotech), but contrary to the reviewer's characterization of the findings, this study shows that blocking internalization **doesn't not** change MAPK signalling. In order to have a balance view, as suggested many times by this reviewer in his/her critiques, we also cited another study showing that blocking V2R internalization totally inhibited MAPK signaling promoted by this receptor (Beautrait et al., Nat. Commun. 2017).

“In that respect we did not observe persistence in AT1R-mediated MAPK signaling with Rasarfin when receptor internalization was inhibited, as reported for other GPCRs (Gosh et al., 2017) and as it would be expected if AT1R continued signaling via G proteins at the PM. For other GPCRs like the V2R, however, inhibition of internalization was sufficient to totally block ERK1/2 activation (Beautrait et al., 2017)”.

8. Overall, looking at the entire paper again, there is much improvement after the revision but much remains to be improved and clarified based on the new additions.

We feel that we have addressed the points raised by this reviewer and present a balance view of the existing literature. We hope that the present modifications will satisfy this reviewer.

Reviewer #2 (Remarks to the Author):

“Initially I had two main issues with this manuscript, MD simulations and quality of docking results (the latter was more about reporting issue than actually how docking was carried out). These issues are now solved”

We thank this reviewer for his comments and appreciation of the improvements made to the manuscript